# Response of carbon cycle to drier conditions in the mid-Holocene in central China

Xianyu Huang[1,2], Richard D. Pancost[3], Jiantao Xue[2], Yansheng Gu[1], Richard P. Evershed[3] & Shucheng Xie[1,2]

The nature and extent to which hydrological changes induced by the Asian summer monsoon affected key biogeochemical processes remain poorly defined. This study explores the relationship between peatland drying and carbon cycling on centennial timescales in central China using lipid biomarkers. The difference between peat $n$-alkane $\delta^2H$ and a nearby stalagmite $\delta^{18}O$ record reveals that intervals of prominent peatland drying occurred during the mid-Holocene. Synchronous with these drier conditions, leaf wax $\delta^{13}C$ values show large negative excursions, with the utilization of $CO_2$ respired from the peatland subsurface for plant photosynthesis being a possible mechanism. Crucially, successive drying events appear to have had a cumulative impact on the susceptibility of peat carbon stores to climate change. Concurrently, bacterially derived hopane $\delta^{13}C$ values suggest the occurrence of enhanced methane oxidation during the drier periods. Collectively, these observations expand our understanding of how respiration and degradation of peat are enhanced during drying events.

[1] State Key Laboratory of Biogeology and Environmental Geology, China University of Geosciences, Wuhan 430074, P.R. China. [2] Laboratory of Critical Zone Evolution, School of Earth Sciences, China University of Geosciences, Wuhan 430074, P.R. China. [3] Organic Geochemistry Unit, Cabot Institute and School of Chemistry, University of Bristol, Cantock's Close, Bristol BS8 1TS, UK. Correspondence and requests for materials should be addressed to S.X. (email: xiecug@163.com)

Peatlands are a vast store of organic carbon and play a significant role in the global carbon cycle[1, 2]. The height of the water table in such environments is the primary influence on carbon degradation pathways and peatland carbon storage, exerting local control on redox conditions in the shallow subsurface[3]. Hence, changes in peatland hydrology will impact carbon storage; for example, dry conditions associated with drought bring about depression of the water table, enhancing degradation of organic matter and release of $CO_2$ to the atmosphere[3]. Because global warming is likely to produce more frequent and/or severe droughts in many regions[4], including in the monsoon region of China, it is crucial to better understand the relationship between hydrological change and the peatland carbon cycle across multiple timescales and particularly in regions expected to experience changing rainfall under future warming scenarios.

The hydrological impact on peatland carbon cycles, especially on varying timescales, continues to be debated. Even on annual timescales, the relationship between water-table lowering and soil organic carbon (SOC) dynamics in peatlands remains unclear; the conventional viewpoint is that drier conditions enhance SOC decomposition through the "enzyme latch" mechanism[5]. In contrast, recent work proposed a new "iron gate" mechanism to interpret the negative relationship between water-table decline and SOC decomposition in settings with abundant iron[6]. On longer timescales, our knowledge of how the peatland carbon cycle responds to drier conditions is even poorer, particularly in monsoon dominated regions[7, 8]. In East Asia, monsoon-mediated rainfall has varied both spatially and temporally since the late deglaciation[9, 10], likely impacting the carbon cycle[8].

In this study, we examine the response of the carbon cycle in a central China peatland to hydrological change over the past 18 ky, but especially to dry intervals during the middle Holocene. During the mid-Holocene, pronounced drier conditions occurred commonly in Indian monsoon dominated regions[11]. In eastern China, however, evidence for drier conditions during the mid-Holocene is limited, with most evidence coming from central China[10, 12]. The occurrence of such conditions in the middle Holocene in central China provides an opportunity to explore the relationship between paleohydrological conditions and the peatland carbon cycle.

The Dajiuhu peatland is a typical subtropical subalpine peatland in central China (Fig. 1)[13], and due to the monsoon-influenced climate, it was exposed to severe drying and flooding events[12]. To reconstruct paleohydrological conditions, we determined the controls on hydrogen isotope compositions in modern

pore water, plants, and peats; based on those constraints and radiocarbon chronology, we use the difference between Dajiuhu leaf wax hydrogen isotope compositions ($\delta^2H_{wax}$) and the nearby Sanbao stalagmite $\delta^{18}O$ sequence[14], complemented by other biomarker indicators of vegetation and bacterial changes, to identify potential dry intervals in the Dajiuhu sequence over the past 18 ky. These records are then integrated with $\delta^{13}C$ values for the same leaf waxes, as well as those of bacterially derived hopanes and carbon accumulation rates, to explore the response of peatland carbon cycle to drier conditions in the middle Holocene. Collectively, this study shows that the peatland carbon cycle is strongly sensitive to paleohydrological changes, expanding our understanding of how respiration and degradation of peat are enhanced during dry intervals.

## Results

### $\delta^2H$ values of pore water and surface peat and modern plant lipids.

The 1-year monitoring of pore water $\delta^2H$ ($\delta^2H_{pw}$) values in Dajiuhu reveals a clear depth pattern (Fig. 2). The $\delta^2H_{pw}$ values in the surface 30 cm are highly variable, whereas the $\delta^2H_{pw}$ values are stable at depths from 50 to 160 cm throughout the year. In 2015, the mean $\delta^2H_{pw}$ value in the upper 30 cm ($-46‰$, $n = 50$) was significantly different ($t$-test, $p < 0.0001$) from the mean $\delta^2H_{pw}$ value for 50–160 cm ($-56‰$; $n = 69$). Moreover, both the shallow and deep pore water mean $\delta^2H_{pw}$ values were enriched relative to the yearly averaged $\delta^2H$ value ($-64‰$) of Dajiuhu precipitation derived from the model of Bowen et al.[15]. Modeled annual $\delta^2H$ values of precipitation must be considered cautiously when applied to a specific site, especially at higher altitude locations; however, the $\delta^2H$ values of precipitation measured in June and July 2015 (Supplementary Table 1) are similar or even lower (avg. $-82‰$) than those estimated from models. Such a difference between the mean $\delta^2H_{pw}$ and the actual rainfall $\delta D$ data clearly indicates that the $\delta^2H$ values of pore water are affected by evaporation.

Peat-forming plants uptake peat water as the hydrogen source for lipid biosynthesis. Previous studies have shown that leaf wax $\delta^2H$ values can be affected by various physiological and environmental factors, such as plant life forms, leaf wax production time and regeneration rate, and evapotranspiration[16, 17]. Leaf samples of dominant herb species (*Carex argyi*, *Sanguisorba officinalis*, *Euphorbia esula*) in Dajiuhu exhibited a mean $n$-$C_{29}$ $\delta^2H$ ($\delta^2H_{29}$) value of $-198‰$ and a mean $n$-$C_{31}$ $\delta^2H$ value of $-191‰$ during the mature stage in 2010 (July–September). The root depths of these herb species range from 12 to 29 cm ($n = 50$ for each species). Assuming the mean

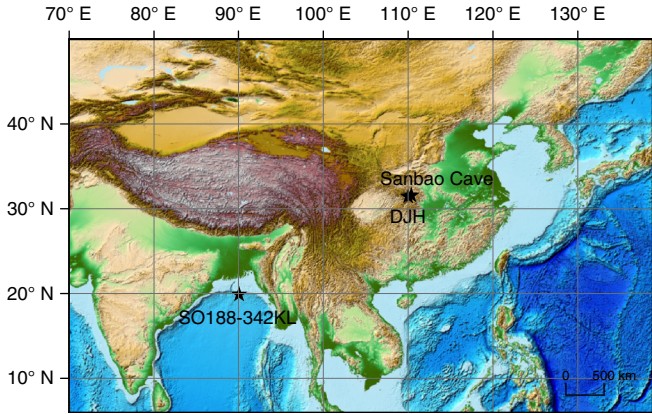

**Fig. 1** Map of the sampling site. The Sanbao Cave and SO188-342KL site are also labeled. The public ETOPO1 data downloaded from https://ngdc.noaa.gov/mgg/global/relief/ETOPO1/image/ were used to plot the topographic map with the software of ArcGIS 9

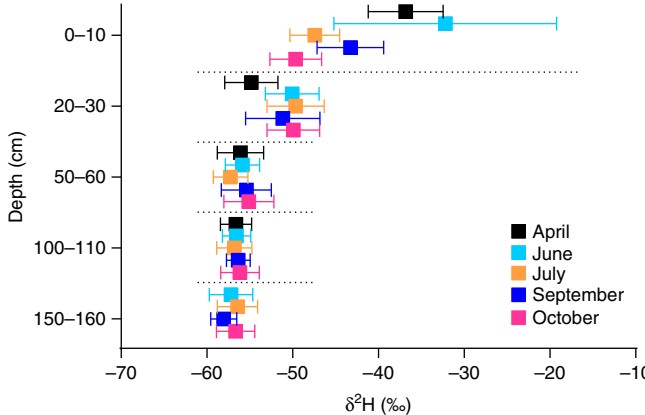

**Fig. 2** Seasonal fluctuations of $\delta^2H$ values in peat pore water in 2015. For each depth at a sampling time, five repeat water samples were collected. Error bars represent 1 s.d. of the batch samples ($n = 5$)

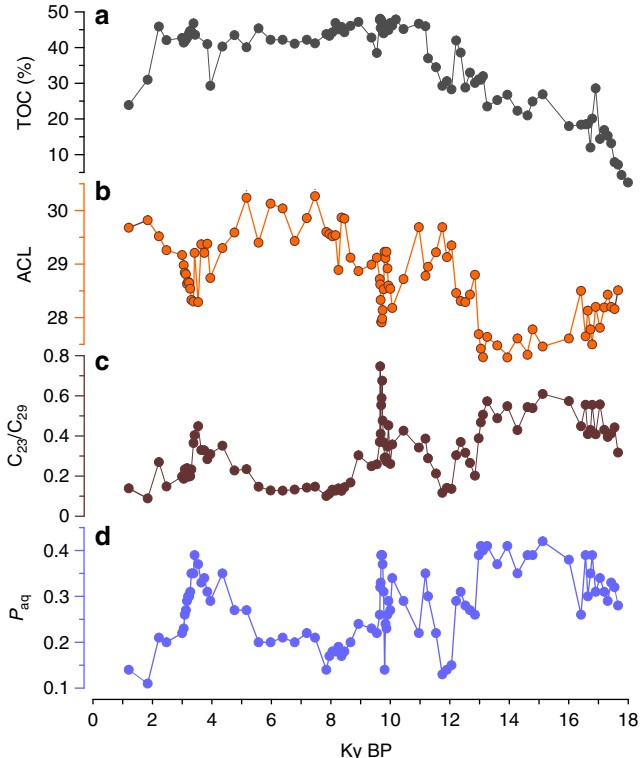

**Fig. 3** Variations of TOC and *n*-alkane ratios in the ZK-5 peat core. **a** TOC. **b** ACL. **c** $C_{23}/C_{29}$ ratio. **d** $P_{aq}$

$\delta^2 H_{pw}$ value of the surface 30 cm represents the annual average, the hydrogen isotope fractionation from the pore water to leaf wax *n*-$C_{29}$ alkane ($\varepsilon_{alk/p}$) is −159‰. Such a $\varepsilon_{alk/p}$ value is larger than a previously calculated value based on surface soil samples in eastern China (−130‰ to −140‰)[18], and the biosynthesis value of forbs (−113‰ ± 31‰)[16]. These differences could result from the response of $\varepsilon_{alk/p}$ to plant habitat conditions (e.g., relative humidity and its impact on evapotranspiration), seasonality of leaf wax production[19], or differences among plant species[16].

Peat deposits always have very high organic matter contents (normally >30%; Fig. 3). In such a terrestrial setting, in situ peat-forming plants contribute almost all long-chain *n*-alkanes to the underlying peat horizons[20, 21]. Consistent with this, in Dajiuhu, the mean $\delta^2 H_{29}$ value (−204‰)[22] in surface peats (*n* = 26) is indistinguishable from those of the plant leaves, indicating an isotopic signal inherited from leaves, without a significant alteration during early diagenesis. Collectively, these observations mean that sedimentary leaf wax $\delta^2 H$ values record those of the peat-forming plants, which in turn appear to be strongly governed by both evaporative and evapotranspirative processes.

**Leaf wax molecular ratios and $\delta^2 H$ and $\delta^{13}C$ values in the peat core**. Long-chain *n*-alkanes, with a strong odd-over-even predominance (carbon preference index (CPI) 2.2–9.4, averaging 5.9), are abundant in all sections of ZK-5 (Supplementary Fig. 1 and Supplementary Table 2). The concentrations of total *n*-alkanes range from 25 to 860 μg g$^{-1}$ dry peat, with an average of 250 μg g$^{-1}$ dry peat (Supplementary Fig. 1). The average chain length (ACL) values range from 27.3 to 30.3 with an average of 28.8 in the whole peat core, with lower values during glacial intervals. The *Sphagnum* associated indices[23], $C_{23}/C_{29}$ (the concentration of $C_{23}$ relative to that of $C_{29}$ *n*-alkane) and $P_{aq}$ (*P*-aqueous ratio), vary closely and exhibit quite low values from 9 to 3.5 ky (Fig. 3).

Since *n*-$C_{29}$ and *n*-$C_{31}$ are the predominant long-chain *n*-alkanes, and their $\delta^2 H$ values vary closely (Supplementary Fig. 2 and Supplementary Table 3), the former is used as a representative of leaf wax *n*-alkanes in the following discussion. Although *n*-$C_{23}$ provides a useful additional record more reflective of *Sphagnum* plants[23], its abundance was low in parts of the profile, preventing generation of a complete record. Over the whole peat core, $\delta^2 H_{29}$ values fluctuate between −188 and −233‰ (Fig. 4). $\delta^2 H_{29}$ values are relatively high during the last glacial, except for the interval from 13.2 to 11.5 ky, and generally lower during the Holocene, which is broadly consistent with other records from the region[24, 25]. After the onset of the Holocene, the $\delta^2 H_{29}$ values are relatively high varying from 11.5 to 10.3 ky and then become lower toward 7 ky. The interval of 7–3.5 ky is characterized by highly variable $\delta^2 H_{29}$ values, with some values being as high as those observed in the glacial. From 3.0 ky onwards, the $\delta^2 H_{29}$ values are relatively low and constant.

In the ZK-5 core, the $\delta^{13}C$ values of *n*-$C_{29}$ ($\delta^{13}C_{29}$) and *n*-$C_{31}$ ($\delta^{13}C_{31}$) alkanes vary between −30.3 and −36.7‰ over the last 18 ky (Fig. 5 and Supplementary Fig. 2). The mean $\delta^{13}C$ values of *n*-$C_{29}$, and *n*-$C_{31}$ are similar and relatively low (−32.7‰ and −32.0‰, respectively), indicating that they primarily derive from $C_3$ plants[26] and are consistent with the dominant flora contributors in peat sequences[27]. Over the last 18 ky, $\delta^{13}C_{29}$ values vary by as much as 6‰, far exceeding the influence of changes in the concentration and carbon isotopic composition of atmospheric $CO_2$ since the Last Glacial Maximum[28]. These $\delta^{13}C_{29}$ variations normally occur rapidly. For example, in the interval from 4.5 to 3.5 ky, the $\delta^{13}C_{29}$ values decrease by 5‰ in 100–200 years. These variations are larger than those observed in other investigations of Holocene peat vegetation[29–31].

**Hopane $\delta^{13}C$ values in the peat core**. Hopane concentrations in the ZK-5 core are high but variable. The distribution is dominated by the 17α,21β(H)-homohopane with an R-configuration at C-22 ($C_{31}$ αβ) (Supplementary Fig. 3), consistent with a previous study in Dajiuhu on a different core (ZK-3)[32] and other investigations of peat deposits[29, 30, 33]. Throughout the whole 18 ky, the $\delta^{13}C$ values of the $C_{31}$ αβ homohopane ($\delta^{13}C_{31\alpha\beta}$) range from −22.5‰ to −30.9‰, which is 5–6‰ higher than those of the long-chain *n*-alkanes (Supplementary Fig. 4 and Supplementary Table 4). These isotopic signatures are consistent with previous reports from other peats[29, 30, 33, 34]. The 17β,21β(H)-norhopane ($C_{29}$ ββ) is also present but less abundant than the 17α,21β(H)-homohopane; it generally has lower and more variable $\delta^{13}C$ values than the $C_{31}$ αβ hopane (Fig. 6). Over the whole 18 ky, the $\delta^{13}C_{31\alpha\beta}$ values generally increase, with relatively higher values occurring in the Holocene than in the deglacial. In contrast, the $\delta^{13}C$ values of $C_{29}$ ββ ($\delta^{13}C_{29\beta\beta}$) do not show any clear temporal trend, but are characterized by lower values from 15 to 11.5 ky and a highly variable interval from 9 to 3 ky (Fig. 5).

**Discussion**

In this study, to explore the relationship between peatland carbon cycle and drier conditions, we combine leaf wax hydrogen and carbon isotope analyses on the same compounds in the same samples, which minimize the influence of other factors, such as vegetation source and sedimentological leads and lags that will affect, for example, microbial biomarkers[8, 35].

Leaf wax $\delta^2 H$ values reveal both changes in water source on glacial–interglacial timescales and in response to Holocene dry intervals. During photosynthesis, terrestrial plants utilize soil water as their major H source, thereby recording the isotopic signatures of the source water, i.e., precipitation ($\delta^2 H_p$). Leaf wax $\delta^2 H$ values are affected by additional factors, such as soil

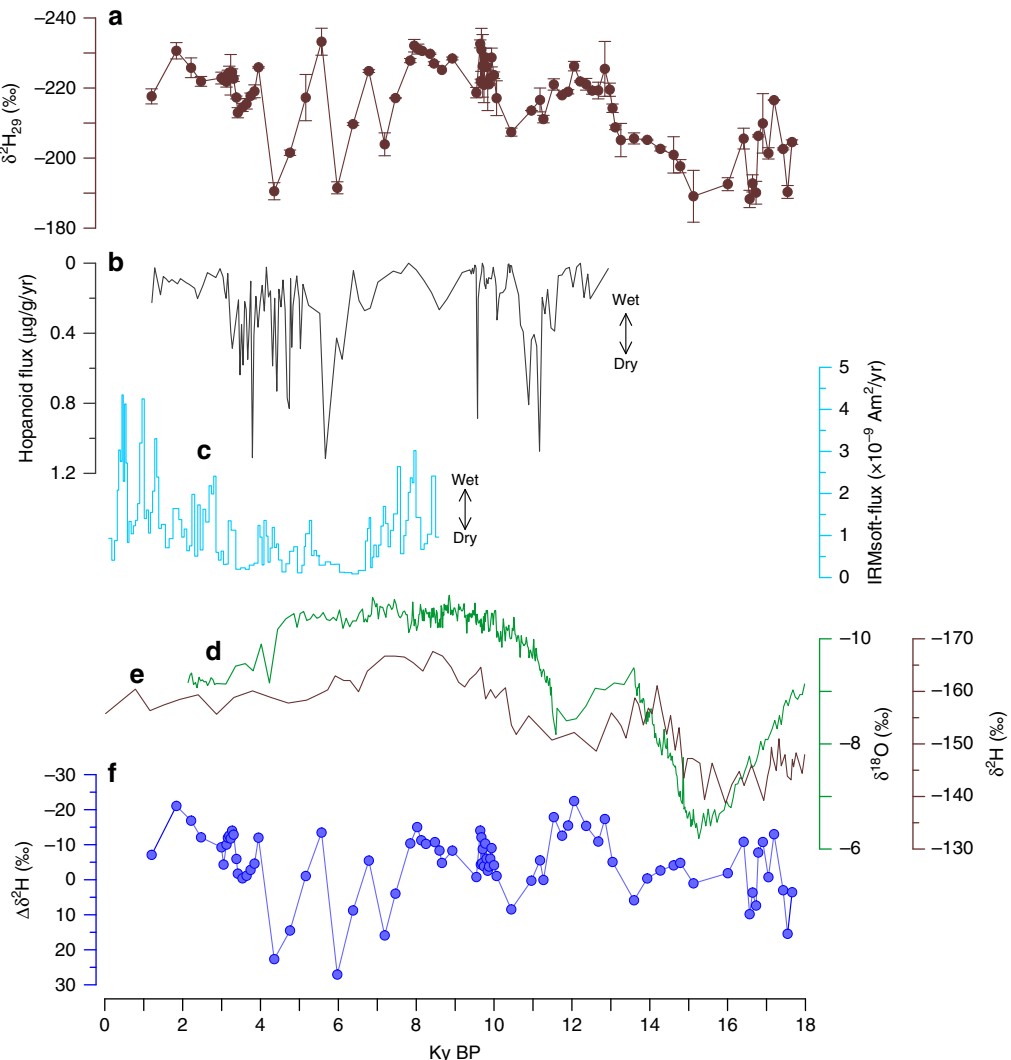

**Fig. 4** Comparisons of paleohydrological records. **a** ZK-5 $\delta^2H_{29}$ values. **b** ZK-3 hopanoid flux[12]. **c** IRM$_{soft\text{-}flux}$ in stalagmite HS4[10]. **d** The calcite $\delta^{18}O$ record from Sanbao Cave[14]. **e** The $C_{29}$ and $C_{31}$ n-alkane weight-averaged $\delta^2H$ values of SO188-342KL[37] over the last 18 ky. **f** The $\Delta\delta^2H$ (between Sanbao Cave carbonate and ZK-5 leaf waxes). Error bars represent 1 s.d. of replicate runs

evaporation, and leaf transpiration, as well as plant-specific physiological and biochemical differences[16]. In peat deposits, in situ peat-forming plants, especially herbaceous subaerial plants ($C_3$ plants), are the predominant contributor of long-chain n-alkanes[35], which has also been confirmed by our previous study at Dajiuhu[36]. Such an inference is further supported by the n-alkane ratios, which reveal a predominance of vascular plants rather than *Sphagnum* during much of the Holocene and especially the last 9 ky (Fig. 3). Moreover, where biomarkers indicate changes in peat-forming plant distributions (Fig. 3 and see below), they are not correlated with leaf wax $\delta^2H$ values and certainly do not appear to be driving variations in those values. This likely reflects the narrower range of sources for the high-molecular-weight leaf waxes, $C_3$ herb plants that generally exhibit a narrow range of apparent hydrogen isotope fractionation between source water and wax lipids[16]. Thus, variations in plant life forms appear to be not an important control on $\delta^2H$ variations in the ZK-5 $\delta^2H_{29}$ sequence.

The broad match ($r = 0.63$, $p < 0.001$) between the ZK-5 $\delta^2H_{29}$ sequence and the nearby Sanbao Cave (<50 km) $\delta^{18}O_{carbonate}$ record[14] is consistent with vapor source being an important factor controlling the ZK-5 $\delta^2H_{29}$ values on millennial timescales (Fig. 4). This inference is further supported by the first order

similarity to the $\delta^2H_{wax}$ record from the Bengal Bay[37], an important vapor source for the East Asia region[38].

However, prominent differences exist between the Sanbao calcite $\delta^{18}O$ and the ZK-5 $\delta^2H_{29}$ records, exemplified by large variations in the $\Delta\delta^2H$ between the two sites (determined by converting the former into meteoric $\delta^2H$ values, Fig. 4), especially during the 7.4–3 ky interval. This indicates that factors other than vapor source control the $\delta^2H_{29}$ variations at Dajiuhu. Relative humidity, via its influence on evaporation and/or evapo-transpiration, is a likely factor[39]. Peats are known to be particularly sensitive to changes in evaporative water balance[40], and therefore, it is unsurprising that the Dajiuhu sequence exhibits variability not observed for the Sanbao Cave (for more detailed discussion, please refer the Supplementary Note 1). This is supported by the modern vertical profile of $\delta^2H_p$ values, in which the upper layers are seasonally variable and $^2H$-enriched relative to deeper sections (Fig. 2) and precipitation water, presumably due to evaporative enrichment.

An interval of relatively drier conditions in the mid-Holocene Dajiuhu sequence is further supported by biomarker indicators of vegetation (Fig. 3). Leaf wax indicators of peat vegetation change must be used with caution[41], but in the Dajiuhu peat sequence they exhibit significant changes that could reflect changes in the

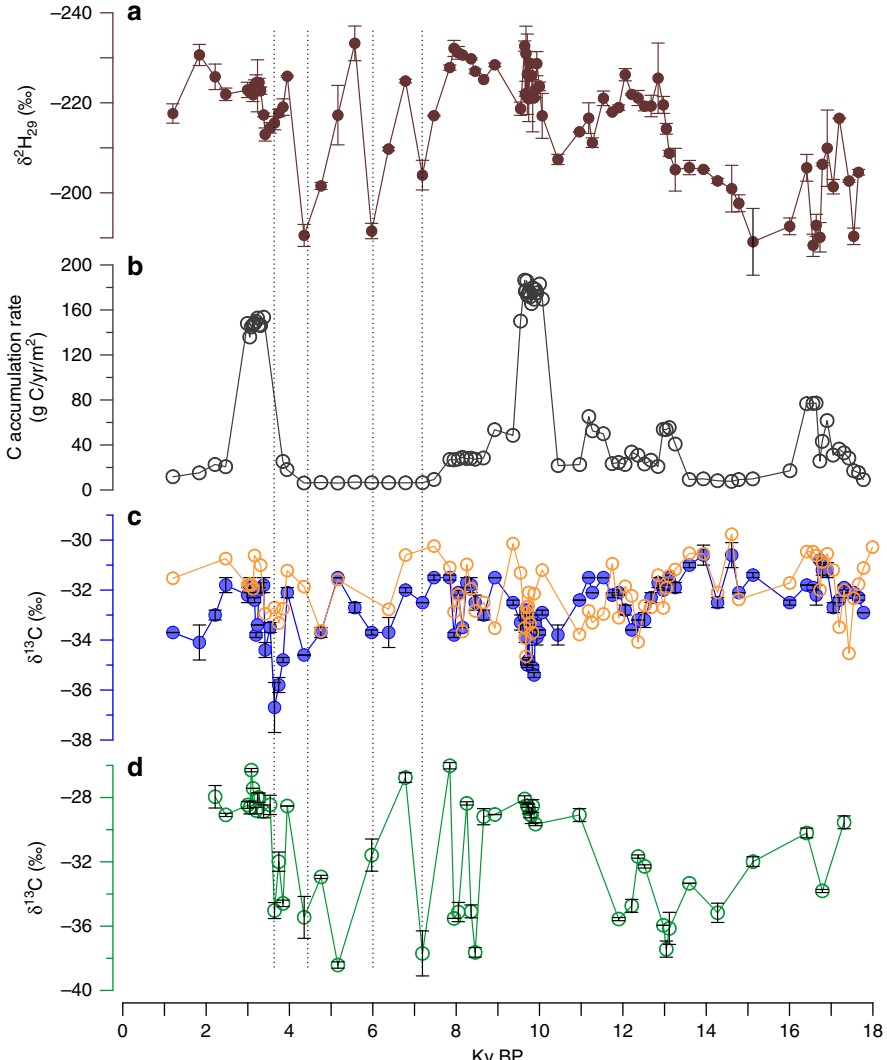

**Fig. 5** Comparisons of paleohydrological and carbon-related records in ZK-5 core. **a** $\delta^2H_{29}$ values. **b** Carbon accumulation rate. **c** The $\delta^{13}C$ values of $C_{23}$ (unfilled circle) and $C_{29}$ n-alkane (filled circle). **d** $C_{29}$ $\beta\beta$ hopane. Error bars represent 1 s.d. of replicate runs. The dashed vertical lines infer the drier episodes

relative importance of *Sphagnum* species (high $n\text{-}C_{23}/n\text{-}C_{29}$ ratios and high $P_{aq}$ indices), which tend to dominate under wetter conditions[23]. Both indices are low over the past 9 ky and especially from 9 to 5 ky. This corresponds with but slightly precedes $\delta^2H$ evidence for dry conditions, suggesting different climatic thresholds for vegetation change. Additional evidence for mid-Holocene aridity is provided by hopanoid abundances in the adjacent ZK-3 core (Supplementary Fig. 5). In the Dajiuhu peatland, hopanoids are mainly biosynthesized by aerobic bacteria, such that hopanoid concentrations serve as a proxy for water-table depth[12], i.e., high-hopanoid abundances are indicative of a deeper water table and aerobic conditions. High abundances are associated with high $\Delta\delta^2H_{29}$ values and low inferred relative humidity (Fig. 4), and both proxies indicate prolonged drier conditions at 11.6–10.6 ky, and 7–3 ky.

It is unclear if mid-Holocene drier conditions in the Dajiuhu peat sequence reflect a regional climate event and again we note that they are not documented in the Sanbao calcite $\delta^{18}O$ record. However, a prolonged drying, and perhaps even drought, during the mid-Holocene has also been inferred from the $IRM_{soft\text{-}flux}$ in speleothems of central China[10]. This proxy records the flux of soil-derived magnetic minerals and

correlates with rainfall amount and intensity, and in particular ENSO-related storms. During 6.7–3.4 ky, $IRM_{soft\text{-}flux}$ exhibits lower values, suggesting drier conditions[10]. A synthesis of paleoenvironmental investigations in the Poyang Basin, central China, also demonstrates drier conditions during 6.0–3.6 ka[42]. Modeling further supports the conclusion that drier conditions prevailed in central China during the mid-Holocene[43, 44]. A review of the mid-Holocene dry climate was recently presented in Liu et al.[45].

The drier conditions during the mid-Holocene in central China contrast with the proposed wet interval of 8–3 ky in the north and the south of China[11]. A previous study interpreted such a spatial pattern as the influence of the western Pacific subtropical high (WPSH) and the associated ENSO variance[11]. During the mid-Holocene, the west–east surface sea temperature gradient was strong[46], and thus the average position of WPSH would have moved north and west, such that the middle and lower reaches of the Yangtze River became dominated by downdraft. In fact, a similar mechanism has been proposed to interpret the negative relationship between precipitation in the middle and lower reaches of the Yangtze River and the summer monsoon intensity on decadal timescales[47].

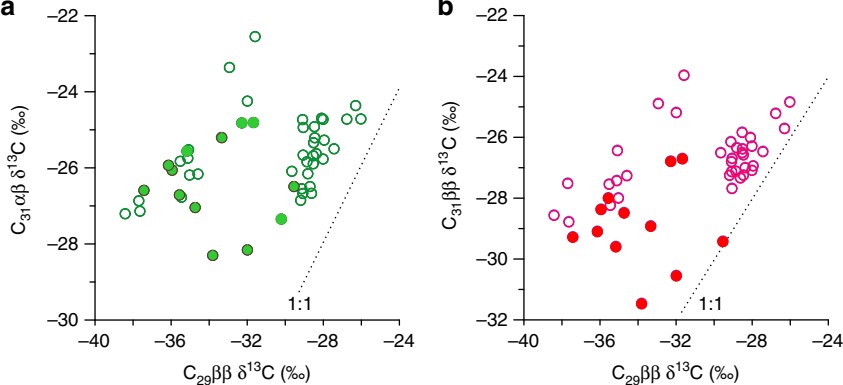

**Fig. 6** Cross-plots between the carbon isotope ratios of hopane homologues. **a** $C_{29}$ $\beta\beta$ vs. $C_{31}$ $\alpha\beta$ hopane. **b** $C_{29}$ $\beta\beta$ vs $C_{31}$ $\beta\beta$ hopane. In each panel, samples from the glacial period are labeled as solid circles, while samples from the Holocene are labeled as open circles

In the middle Holocene, multiple strong but brief positive $\Delta\delta^2H$ shifts occurred (Fig. 4). The three most prominent $\delta^2H$ intervals (20–40‰), centered at 7.2, 5.6, and 4.4 ky, correspond to high-hopanoid concentrations and inferred drier conditions[12] (Fig. 4). The occurrence of prominent drier conditions in the middle Holocene provides an opportunity to explore the relationship between paleohydrological conditions and the peatland carbon cycle. The most direct evidence for this impact is documented by prolonged low carbon accumulation rates from about 9 to 3.5 ky (Fig. 5) in the ZK-5 core. This likely reflects a combination of decreased production and increased respiration, collectively leading to decreased carbon storage. The leaf wax and bacterial biomarker carbon isotopic signatures provide additional insights into changes in carbon cycling during the mid-Holocene dry interval. For $C_3$ plants, leaf $\delta^{13}C$ values are mainly governed by the air isotopic composition and isotope discrimination during photosynthesis ($\varepsilon_p$)[48]. Over the last 18 ky, atmospheric $CO_2$ $\delta^{13}C$ has changed by <1‰[28], far too low to account for the shifts observed here. Moreover, although $CO_2$ concentrations have changed slightly and these affect $\varepsilon_p$ values, the effect is likely to have been small[49], especially during the Holocene. Similarly, $\varepsilon_p$ is sensitive to water stress with dry periods associated with decreased $\varepsilon_p$ and, therefore, high plant $\delta^{13}C$ values[50]. It is unexpected, therefore, that during the drier episodes of the mid-Holocene, leaf wax $\delta^{13}C$ values always display negative excursions (Fig. 5). In addition, vegetation shifts inferred from palynogical data[51] or leaf wax distributions (Fig. 3) do not match the $\delta^{13}C_{29}$ variations.

Consequently, we argue that some of the large shifts in leaf wax $\delta^{13}C$ values record changes in peatland carbon cycling, in particular increased plant uptake of respired $CO_2$ during dry intervals. Dry intervals can be associated with enhanced peat degradation[5], and previous studies have proposed that refixation of microbially respired $CO_2$ is an important mechanism to maintain the higher primary productivity in peat bogs[52–54]. Such respired $CO_2$ will have $\delta^{13}C$ values close to the $\delta^{13}C$ values of bulk OM[55] and much lower than that of atmospheric $CO_2$[56]. Each of the three Holocene positive $\Delta\delta^2H$ shifts is associated with a decrease in leaf wax $\delta^{13}C$ values. Increased microbial respiration under warmer and drier climate conditions, by increasing the release of $^{13}C$-depleted $CO_2$ available for photosynthesis[56, 57], provides a mechanistic link between these observations. This is also consistent with the lower TOC contents and peat accumulation rates from 9 to 3.5 ky (Fig. 5). However, not all episodes of $^{13}C$ depletion are associated with inferred drier intervals, indicating that the relationship between the two was likely nonlinear and/or that other factors govern some of the $\delta^{13}C$ variability in the Dajiuhu peat.

Throughout the interval of inferred dry (but variable) conditions in the mid-Holocene, the amplitudes of the negative $\delta^{13}C_{alk}$ excursions become larger over time; consequently, a relatively minor positive $\Delta\delta^2H$ excursion at 3.4 ky is associated with a large 4.6‰ $\delta^{13}C_{alk}$ shift (Fig. 5). Such a pattern suggests that the supply of respired $CO_2$ could be sensitive to the cumulative effect of drying cycles, rather than simply linearly responding to a single event. Fenner and Freeman[5] observed an increase of carbon losses from peat exposed to climatic variation and suggested that severe drying and subsequent rewetting would destabilize peatland carbon stocks. More contemporary studies emphasize the importance of drying–rewetting on peatland carbon dynamics[58]. As such, the nonlinear coupling of leaf wax carbon and hydrogen isotope ratios suggests that multiple drying cycles led to destabilization of Dajiuhu peat stock and pulses of organic matter respiration on centennial–millennial scales.

The influence of drier conditions on peatland carbon cycling is further evidenced by the carbon isotope shifts of bacterially derived hopanes. The controls on hopane carbon isotopic compositions are complex[33], but previous work on peats suggests that the balance between heterotrophy (including the carbon isotopic composition of different substrates) and methanotrophy is crucial[59]. This is likely true in Dajiuhu, where hopanoids are mainly biosynthesized by aerobic bacteria (based on analyses of sqhC genes)[12, 60]. However, the $C_{29}$ $\beta\beta$ hopane is both more $^{13}C$-depleted and more isotopically variable than the two $C_{31}$ hopane isomers (Fig. 6 and Supplementary Fig. 4).

Previous studies reveal that the carbon isotope offset between $C_{31}$ $\alpha\beta$ homohopane and leaf wax $n$-alkanes is commonly between 4 and 6‰[29]. Consistent with this, in the modern surface peats collected from Dajiuhu in July 2012, the $C_{31}$ $\alpha\beta$ homohopane is 6–7‰ enriched relative to the $C_{29}$ $n$-alkane. This offset suggests that hopane-producing bacteria in acidic peats are probably heterotrophic and utilizing isotopically heavy carbohydrates as their major substrate[29, 33]. However, from the late deglacial to the late Holocene, the $\delta^{13}C_{31\alpha\beta}$ values increase by 2–3‰ (Supplementary Fig. 4), whereas those of the leaf waxes decrease. This isotopic decoupling between the putative organic matter source and the consumer bacteria could arise from a number of factors, but we tentatively propose that it reflects the temperature control on substrate availability. Under cooler glacial conditions, we suggest that lower rates of respiration are associated with a reduced bias toward microbial assimilation of carbohydrates.

In contrast, $\delta^{13}C$ values of the $C_{29}$ $\beta\beta$ hopane, while also being lower in the glacial interval than the Holocene, exhibit a depth profile dominated by profound variability from 9 to 3 ky (in fact,

the $C_{31}$ hopanes also exhibit stronger variability during this interval, although less pronounced than for the $C_{29}$ hopane). The dramatic variability appears to be broadly associated with the drier interval, with $\delta^{13}C_{29\beta\beta}$ values decreasing by up to 10‰, to values as low as −40‰ (Fig. 5). The interval of lowest $\delta^{13}C_{29\beta\beta}$ values is deeper than the drier interval and individual shifts are not directly correlated to shifts in $\Delta\delta^2H$; this is likely due to depth offset, due to bacteria living in subsurface layers. This would be especially true for transient dry events, which would allow hopanoid-producing bacteria to periodically thrive in aerated regions of deeper peat. Consequently, hopane carbon isotope signals appear to stratigraphically lead the dry events recorded by $\Delta\delta^2H$.

Shifts to such low $\delta^{13}C_{29\beta\beta}$ values are difficult to explain via changes in organic matter sources (and in fact, in some cases, $\delta^{13}C_{29\beta\beta}$ values are 4–6‰ lower than those of the n-alkanes), and instead likely reveal contributions from $^{13}C$-depleted methanotrophic bacteria during dry intervals. An increased methanotroph contribution during a dry interval seems counter-intuitive, as a low water table is likely associated with reduced rates of methanogenesis; however, it is similar to findings from a recent study of the Hongyuan peat sequence, southwest China, which revealed very low diploptene $\delta^{13}C$ values during a dry interval of the mid-Holocene[8]. In that work, the low $\delta^{13}C$ values were attributed to more diffusive flux of methane during dry intervals (as opposed to root-mediated transport), which could have facilitated growth of methanotrophs. Alternatively, the low $\delta^{13}C_{29\beta\beta}$ values could arise from changes in the abundance of Sphagnum symbiotic methanotrophs[61], which are known to produce <$C_{31}$ hopenes[62]. However, such a possibility is not supported by the n-alkane ratios during the drier interval in the mid-Holocene, which reveal a low contribution from Sphagnum (Fig. 3). In addition, symbiotic methanotrophs associated with Sphagnum are more active in wetter conditions[63].

By integrating multiple isotope records, this work provides new evidence for Chinese drier intervals on centennial–millennial timescales, as well as direct evidence that these drier conditions impacted the peatland carbon cycle. The difference between peat leaf wax $\delta^2H$ values and the nearby cave calcite $\delta^{18}O$ record reveals that prominent drier intervals, centered at 7.2, 5.6, 4.4, and 3.4 ky, occurred during the mid-Holocene in central China. This conclusion is reinforced by an absence of Sphagnum species during this interval and elevated abundances of hopanoids of putative aerobic bacteria origin. Corresponding to these drier intervals, carbon accumulation rates are very low and leaf wax $\delta^{13}C$ values decrease markedly, opposite to the expected effect of decreased moisture, suggesting an increase in photosynthetic assimilation by the bog vegetation of isotopically depleted $CO_2$ derived from microbial respiration within the peat. The magnitude of the leaf wax $\delta^{13}C$ perturbations increases with successive drying cycles, indicating a cumulative effect of drier conditions on peatland carbon dynamics. At approximately the same time, carbon isotope ratios of bacterial biomarkers, especially those of $C_{29}$ $\beta\beta$ hopane, become much more variable, providing further evidence of a perturbed carbon cycle, which we attribute to changes in the dynamics of methane production, flux, and consumption. Collectively, these processes resulted in a dramatic reduction in carbon accumulation rates, such that this work directly demonstrates that the peatland carbon cycle is sensitive to paleohydrological changes on long-term, centennial to millennial timescales.

## Methods

**Site description.** Dajiuhu is a closed subalpine basin located in the middle reaches of the Yangtze River, central China. This basin has a mean elevation of 1730 m and a total area of 16 km². Since the late deglaciation, peat developed in this basin to a

depth of 2–3 m. The modern dominant peat-forming plants include sedge species, S. officinalis, and Sphagnum palustre. Water in this basin is drained through sinkholes to the Du River. Climate in this region is dominated by the Asian monsoon, with hot-wet summers and cold-dry winters, mean annual precipitation of 1560 mm and mean annual temperature of 7.2 °C. This region is located at the transition from the eastern lowland to the western highland, making it particularly sensitive to climate changes.

A 3-m core (ZK-5; 31°28′56″ N, 109°59′56″ E, a.s.l. 1758 m) was collected from Dajiuhu peatland in July 2013. The upper 2.6-m of ZK-5 core was peat, and the lower section was gray clay, presumably deposited under lacustrine conditions. Sub-samples of peat for biomarker analyses were collected from the upper 2.6 m peat layer, sliced at 1-cm intervals in the field.

**AMS $^{14}C$ dating and chronology.** The chronology of ZK-5 is based on the $^{14}C$ accelerator mass spectrometer (AMS) analyses of 20 organic sediments (Supplementary Table 5). AMS measurements were conducted at Beta AMS Lab (Miami, USA). Following the study of Zhou et al.[64], the 90–300 μm fraction was sieved from the bulk samples, and then subjected to an acid–alkali–acid treatment before AMS analysis. The calendar age was calibrated using the clam age-depth model[65] (Supplementary Fig. 6).

**Lipid extraction and analysis.** Freeze-dried peat samples were ground to pass a 60-mesh sieve (0.18 mm) and were ultrasonically extracted 6 × 10 min with dichloromethane/methanol (9:1, v/v). The apolar fraction was isolated by elution from a silica gel chromatographic column with hexane. Gas chromatography–mass spectrometry analysis of the apolar fraction was conducted with an Agilent 6890 gas chromatograph interfaced with an Agilent 5973 mass selective detector, with the instrumental conditions identical with the previous study[32]. Compound-specific hydrogen isotope compositions of n-alkanes were determined using a Trace GC coupled with a Delta V advantage isotope ratio mass spectrometer. To check the system stability, an n-alkane mixture (n-$C_{23}$, n-$C_{25}$, n-$C_{27}$, n-$C_{29}$, and n-$C_{31}$ alkane) and the Indiana A4 mixture with known $\delta^2H$ values were analyzed between every two samples. Squalane ($\delta^2H$ −167‰) was used as the internal standard. Standard deviation for hydrogen isotope analysis was better than ±5‰, based on at least duplicate analyses. Results are reported in the delta notation (‰) relative to the Vienna Standard Mean Ocean Water standard.

Compound-specific carbon isotope analysis was conducted using a Finnigan Trace GC attached to a Finnigan Delta Plus XP isotope ratio mass spectrometer, equipped with a DB-5MS capillary column (30 m × 0.25 mm × 0.25 μm). The injector temperature was set at 300 °C. The GC oven temperature initiated at 50 °C (held 1 min), and then ramped to 220 °C at a rate of 10 °C min⁻¹ (held 2 min), and further ramped to 300 °C at a rate of 2 °C min⁻¹, and finally to 310 °C at a rate of 10 °C min⁻¹ (held 20 min). Helium was used as the carrier gas (1.4 ml min⁻¹). The combustion oven was set at 950 °C. Instrument performance was verified before and after each sample run using an n-alkane standard mixture with known $\delta^{13}C$ values (n-$C_{16}$–n-$C_{30}$, Indiana University). Reproducibility for specific compounds was better than ±0.5‰ (standard deviation), based on at least duplicate analyses. Results are reported in the delta notation (‰) relative to the VPDB standard.

**Calculations of the n-alkane ratios.** The ACL, CPI, and $P_{aq}$ were calculated using the following equations:

$$ACL = \frac{21 \times C_{21} + 23 \times C_{23} + 25 \times C_{25} + 27 \times C_{27} + 29 \times C_{29} + 31 \times C_{31} + 33 \times C_{33}}{C_{21} + C_{23} + C_{25} + C_{27} + C_{29} + C_{31} + C_{33}}.$$

(1)

$$CPI = \frac{(C_{21} + C_{23} + C_{25} + C_{27} + C_{29} + C_{31}) + (C_{23} + C_{25} + C_{27} + C_{29} + C_{31} + C_{33})}{C_{22} + C_{24} + C_{26} + C_{28} + C_{30} + C_{32}} \times \frac{1}{2}.$$

(2)

$$P_{aq} = \frac{C_{23} + C_{25}}{C_{23} + C_{25} + C_{29} + C_{31}}.$$

(3)

**Sampling of water and $\delta^2H$ analysis.** Peat pore waters were obtained from Dajiuhu during five sampling trips in 2015. These samples were collected using a MacroRhizon soil moisture sampler (with a length of 10 cm and pore size of 0.2 μm; Rhizosphere Research Products B.V., The Netherlands). The samplers were established at five different depths (0–10, 20–30, 50–60, 100–110, and 150–160 cm (except April, for which 150–160 cm was not collected)) at five locations. During June and July 2015, rainfall samples were collected using a 1000-ml flask, blocked with a ping-pong ball on the bottleneck. During water collection, olefin was added to prevent water evaporation.

The hydrogen isotopic compositions of the peat pore waters and rainfall were analyzed using an IWA-35-EP Liquid Water Isotope Analyzer (LGR, USA) at the State Key Laboratory of Biogeology and Environmental Geology. The analytical precision was better than 0.2‰ for $\delta^{18}O$ and 0.6‰ for $\delta^2H$. Data for the batch of water samples collected in October 2015 have been published in Huang et al.[22].

**Total organic carbon concentration analysis**. The peat samples were first freeze-dried and then ground to fine powder (<100 mesh) and homogenized. The total organic carbon concentration was measured on the Vario MICRO cube Element Analyzer.

**Calculation of carbon accumulation rate**. The method was followed reference[7]:

C accumulation rate (g C m$^{-2}$ year$^{-1}$) = peat accumulation rate (mm year$^{-1}$)/1000 × ash-free bulk density (g m$^{-3}$) × 0.52 × TOC (g g$^{-1}$ dry weight)  (4)

The ash-free bulk density utilized the mean value (0.12 g cm$^{-3}$) from a nearby peat core retrieved from Dajiuhu[66].

**Calculation of Δδ$^2$H**. The Δδ$^2$H values were calculated using Eqs. (5) and (6):

$$\delta^2H_p = 7.9 \times \delta^{18}O_{ca} + 8.2,  (5)$$

where δ$^{18}$O$_{ca}$ is the carbonate oxygen isotope values from Sanbao Cave[11]. Here we assume the carbonate δ$^{18}$O record in Sanbao Cave is representative of precipitation oxygen isotope composition, and omit the combined effect of air temperature on precipitation δ$^{18}$O values and of cave temperature on oxygen isotope fractionation during carbonate precipitation from drip water (ca. +0.05‰ °C$^{-1}$ in the eastern China[67]). The intercept (8.2) and slope (7.9) arise from the regional meteoric water line[68]. We then calculated Δδ$^2$H using the equation:

$$\Delta\delta^2H = \delta^2H_{29} - \varepsilon_{alk/w} - \delta^2H_p - a  (6)$$

where ε$_{alk/w}$ is the hydrogen isotope fractionation between long-chain n-alkanes and precipitation, assumed to be −149‰ based on our recent study of surface peat in Dajiuhu[22]. The constant a represents the altitude effect on precipitation oxygen isotopes (−0.021‰ m$^{-1}$ for Shennongjia Mountain[69]), applied to the 200 m altitude difference between Sanbao Cave and Dajiuhu.

**Data availability**. The data that support the findings of this study are included in the supplementary information files.

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

## Acknowledgements

This work was supported by the NSFC (41330103, 41472308), 111 Project (grant B08030), and the fundamental research funds for the central universities (CUGCJ1703). Z. Zhang from Shennongjia National Park Administration, Dr. X. Chen, Y. Gao, R. Wang, Y. Zhang, B. Zhao, Q. Song are thanked for their help in the field. Dr. X. Li and Y. Wang are thanked for the analysis of peat water δ2H composition.

## Author contributions

X.H., J.X., and Y.G. performed the field work; J.X. and X.H. conducted biomarker and water isotope composition measurement and analysis. S.X. and X.H. designed the project. The manuscript was written by X.H., R.D.P., R.P.E. and S.X. with contributions from all authors.

## Additional information

**Competing interests:** The authors declare no competing interests.

