## [Peer Review File · Nature Communications]

Reviewers' comments:

Reviewer #1 (Remarks to the Author):

Huang and his colleagues interweave a suite of organic geochemical paleoenvironmental proxies in this elegant and interesting contribution. Their goal is to explore for evidence that some of the abundant amounts of organic carbon sequestered in peatlands can be released and recycled and to identify the situations when this process might occur. To this end, they combine compound-specific carbon and hydrogen isotope measurements of leaf-wax components that represent the principal plants responsible for peat accumulation with compound-specific carbon isotope measurements of biomarkers for aerobic bacteria that would be responsible for much of the carbon recycling. The hydrogen isotope measurements are indicators of former wet-dry alternations in the Dajiuhu peatland that the authors study. These proxies are complemented by oxygen isotope variations recorded in the nearby Shanbao Cave to allow the authors to reconstruct the mid-Holocene paleohydrologic history of this area. What they conclude is that during episodes of drier climate and depressed water levels, peat was subject to bacterial oxidation that released isotopically light carbon dioxide that was assimilated by the peat-forming plants, yielding more negative carbon isotopic values of their leaf waxes. This chain of events makes biogeochemical sense, yet it has not been so elegantly documented before.

I found the manuscript difficult to fully appreciate on my first reading, but I came to admire how the authors so eloquently explained and described the details of their sophisticated study on my second reading. On a larger scale, the results of their study are significant to understanding what could have happened to the peatlands that presumably existed prior to the last glaciation. Virtually all of them vanished during the cooler and drier climate that prevailed from 30 ka to about 18 ka. The study by Huang et al. gives a very nice example of how the organic carbon interred in preglacial peatlands could have been oxidized, released, and globally redistributed.

The document is written and presented exceptionally well; I found only a few minor editorial improvements that might be made:

Line 38 – replace “deepening” with “depression”

Line 177 – change “particularly” to “particular”

Line 258 – change “magnitude” to “magnitudes”

Phil Meyers
May 16, 2017

Reviewer #2 (Remarks to the Author):

This paper reports hydrogen isotopic values of C₂₉ n-alkane from Da Ju Hu for the past 18 thousand years. The key data is the observation of large dD fluctuations between 8 and 3.5 ka in Da Ju Hu, which is interpreted as major drought events in the region. The overall dD C₂₉ trend in the past 18,000 years however is generally similar to the Shanbao d₁₈O. The large fluctuations in dD values during mid Holocene are interpreted as major decreases in relative humidity, perhaps not associated with decreases in precipitation (this is not clear in the paper). The low d₁₃C excursions of C₂₉ n-alkane during the same period of time are interpreted as more carbon uptake from respired CO₂ from enhanced decomposition in more aerobic conditions of organic matter during dry times. Hopanoids (total) appear to show also larger fluctuations during the mid Holocene, which is also suggested to be related to aerobic bacterial production.

Overall it is an interesting data set, with many unexpected trends and unusual features. It is well possible that the DJH peatland did undergo large changes in hydrology due to various reasons, not all related to climate, but perhaps regional hydrology and sedimentation system. The data are

definitely worth publishing.

However, I feel that there are way too many speculations throughout the paper, including interpretation of isotopic ratios and sources of various compounds etc based on relatively sketchy evidence. I am also puzzled with what the main new points of the paper are. As authors cited the Fenner and Freeman Nature Geosci paper – it is not new that drought leads to carbon loss in peatland. Similarly, authors state a number of papers already found dry mid Holocene in the broad region (line 153-154). What is new for this paper? Is it climate story new or carbon cycle rate new? The title makes it sound like the main emphasis is that drought leads to carbon loss in peatland during the mid Holocene, but this point is hardly a new discovery.

Here are a list of specific comments on the paper:

- Organic carbon contents or loss on ignition data should be plotted and they should be measured to very high resolution since these are easy to measure. The data are crucial for supporting authors' argument that mid Holocene period is a time of repeated drought, resulting in excessive burning of peatland carbon and greatly enhanced microbially respired CO₂.

- This paper focuses on Holocene climate and carbon cycling interpretations, but oddly for the past 8 thousand years, there are only 3 radiocarbon dates, whereas there are 14 dates before 8 ka. Large gaps with no radiocarbon dates from 8 to 3.5 ka, and no dates from 3 kyr bp to present. I can see peat accumulation rate is low, but if authors want to claim a new climate record, it is necessary add more radiocarbon dates rather than relying on interpolation alone. There are extremely large changes in sedimentation rate in the past 8 thousand years, with extremely low rate between 8 and 3.5 ka. It is interesting to see the surge in sedimentation rate between ca 3.5 to 3 ka, but then a dramatic slowdown in accumulation from 3 ka to present. These are extremely important results and need to be carefully explained.

- Fluxes of n-alkanes and various chain length and distribution data should also be plotted at least in the supplementary data. If hopanoids are analyzed throughout the core with high resolution as shown in figure 1, n-alkane data should also be available. If different hopanoids have different sources as proposed, the fluxes should be plotted separately rather than together as total amounts.

- It is hard to understand why the large fluctuation in n-alkane dD values in mid Holocene is not seen in Shabao d18O data. Low relative humidity can increase evaporation of soil waters, which should be seen for cave water isotopic ratios too. Peatland tends to be in places with the most abundant precipitation, it is hard to imagine a larger evaporative isotopic effect in peatland region than where speleothem is formed nearby. Also, during evaporation oxygen isotopic ratios display much greater isotopic effect than hydrogen due to the stronger kinetic isotopic fractionation. A muted oxygen isotope effect along with a large hydrogen isotope change is difficult to explain. If pollen or macrofossil data are available, it is important these data are plotted for comparison. Different plants can have large hydrogen isotopic differences, and it is possible some of the observed hydrogen isotopic fluctuations during the mid Holocene are due to vegetation effects.

- I actually do not see clear correspondence of low n-alkane dD values with enhanced hopanoid fluxes. There are only a few points with relatively high hopanoid fluxes but these do not directly correspond to high n-alkane dD values (indicating drought as argued). I also fail to see how n-alkane high dD values are correlated to hopanoid d13C changes. If authors want to show they are correlated show how, they should plot X-Y plots for the same samples to see if there are any real correspondence.

- Interpreting different hopanoids have different sources based on d13C data alone is quite a stretch. Often there are natural differences in isotopic ratios between different hopanoids, even if they are made by the same organism. Note some hopanoids can be produced by fern as well.

- There is some correlation between δD and $\delta^{13}C$ for C₂₉ n-alkane for mid Holocene drought section but not particularly strong. To say low $\delta^{13}C$ values during the proposed dry phases originates from ambient CO₂ with greater proportion of recycled CO₂ is not convincing to me. If the peat land has stopped accumulating as suggested by extremely low accumulation rate during this time, there would not be so much carbon to burn to produce CO₂ with low $\delta^{13}C$ values. Again there is no TOC data plotted in this paper hence difficult to judge for me. Often it is during the wet times, increased methane production and resulting microbially respired CO₂ from methane with low $\delta^{13}C$ values could contribute low $\delta^{13}C$ values of CO₂, rather than the opposite. Also, as discussed in the paper, dryer conditions can lead to reduced carbon isotopic fractionation of higher plants, and higher $\delta^{13}C$ values which can then counter the proposed source carbon effect. It is complicated.

- Figure 2 needs some dashed lines to separate various depth sections. The number of water samples for each collection season needs to be stated clearly regardless whether the data have been published before. Are there daily water sample collections or weekly collections for example?

- If pronounced dry episodes in mid Holocene are already observed in China, authors should plot these data in this paper to help readers.

Reviewer #3 (Remarks to the Author):

This manuscript presents δ^2H_{wax} and $\delta^{13}C$ data from a peatbog in northern China to identify drought events. The scientific methods and data seem sound and the $\delta^{13}C$ data is interesting. However, there are a few items missing from the manuscript. First, I think this is an important data set, but the authors could bring out the wider implications of it more strongly.

In general, I think the discussion regarding the δ^2H record needs to be clearer. The diversion between Sanbao and Dajiuhu from 8 and 5 ka (broadly speaking) suggesting drought conditions does seem reasonable and important. However, I find the discussion regarding the match with the Bay of Bengal to be confusing and potentially erroneous. Regarding lines 132-137 in particular- are the authors suggesting that vapor source (and thereby wind direction) is a control on δ^2H values? It is certainly surprising that the δ^2H values at Dajiuhu would be similar to the Bay of Bengal δ^2H values given that Dajiuhu is dominated by the EASM and the Bay of Bengal reflects the ISM. While this is not to say that the EASM and ISM cannot be synchronous, increasingly it is becoming clear that they appear to behave asynchronously in the Holocene. This leads me to related point, which is that the authors don't make clear which monsoon system they are referring to.

Lines 66-68- Bowen's model has been shown to have some deficiencies, particularly with regard to higher altitude locations. I would be more comfortable with this interpretation if actual precipitation samples were used to demonstrate this.

Lines 125-160- This is more of a discussion rather than a presentation of results. Move accordingly.

Lines 130-131- What about the possibility that vegetation in this peatland may have been different, particularly during the glacial period?

Lines 138-140- Identify which particular periods you are referring to where there are prominent differences. I see some in Figure 4, but it's not clear if we are looking at the same time frame.

Lines 154-156- Indeed, middle Holocene drought conditions are a prevalent feature of many ISM

records. There are fewer EASM records that show this though and I think this point needs expansion.

Figure 4- Some annotation on the $\Delta\delta^2\text{H}$ panel would be helpful, indicating which axis direction indicates what.

Reviewers' comments:

Response to comments from Reviewer #1:

Huang and his colleagues interweave a suite of organic geochemical paleoenvironmental proxies in this elegant and interesting contribution. Their goal is to explore for evidence that some of the abundant amounts of organic carbon sequestered in peatlands can be released and recycled and to identify the situations when this process might occur. To this end, they combine compound-specific carbon and hydrogen isotope measurements of leaf-wax components that represent the principal plants responsible for peat accumulation with compound-specific carbon isotope measurements of biomarkers for aerobic bacteria that would be responsible for much of the carbon recycling. The hydrogen isotope measurements are indicators of former wet-dry alternations in the Dajihu peatland that the authors study. These proxies are complemented by oxygen isotope variations recorded in the nearby Shanbao Cave to allow the authors to reconstruct the mid-Holocene paleohydrologic history of this area. What they conclude is that during episodes of drier climate and depressed water levels, peat was subject to bacterial oxidation that released isotopically light carbon dioxide that was assimilated by the peat-forming plants, yielding more negative carbon isotopic values of their leaf waxes. This chain of events makes biogeochemical sense, yet it has not been so elegantly documented before.

I found the manuscript difficult to fully appreciate on my first reading, but I came to admire how the authors so eloquently explained and described the details of their sophisticated study on my second reading. On a larger scale, the results of their study are significant to understanding what could have happened to the peatlands that presumably existed prior to the last glaciation. Virtually all of them vanished during the cooler and drier climate that prevailed from 30 ka to about 18 ka. The study by Huang et al. gives a very nice example of how the organic carbon interred in preglacial peatlands could have been oxidized, released, and globally redistributed.

The document is written and presented exceptionally well; I found only a few minor editorial improvements that might be made:

Reply: Thank you very much for the positive comments and for the constructive criticism that has helped improve the manuscript.

Line 38 – replace “deepening” with “depression”

Reply: Accepted.

Line 177 – change “particularly” to “particular”

Reply: Accepted.

Line 258 – change “magnitude” to “magnitudes”

Reply: Accepted.

Response to comments from Reviewer #2:

This paper reports hydrogen isotopic values of C₂₉ n-alkane from Da Ju Hu for the past 18 thousand years. The key data is the observation of large dD fluctuations between 8 and 3.5 ka in Da Ju Hu, which is interpreted as major drought events in the region. The overall dD C₂₉ trend in the past 18,000 years however is generally similar to the Shanbao d₁₈O. The large fluctuations in dD values during mid Holocene are interpreted as major decreases in relative humidity, perhaps not associated with decreases in precipitation (this is not clear in the paper). The low d₁₃C excursions of C₂₉ n-alkane during the same period of time are interpreted as more carbon uptake from respired CO₂ from enhanced decomposition in more aerobic conditions of organic matter during dry times. Hopanoids (total) appear to show also larger fluctuations during the mid Holocene, which is also suggested to be related to aerobic bacterial production.

Overall it is an interesting data set, with many unexpected trends and unusual features. It is well possible that the DJH peatland did undergo large changes in hydrology due to various reasons, not all related to climate, but perhaps regional hydrology and sedimentation system. The data are definitely worth publishing.

However, I feel that there are way too many speculations throughout the paper, including interpretation of isotopic ratios and sources of various compounds etc based on relatively sketchy evidence. I am also puzzled with what the main new points of the paper are. As authors cited the Fenner and Freeman Nature Geosci paper – it is not new that drought leads to carbon loss in peatland. Similarly, authors state a number of papers already found dry mid Holocene in the broad region (line 153-154). What is new for this paper? Is it climate story new or carbon cycle rate new? The title makes it sound like the main emphasis is that drought leads to carbon loss in peatland during the mid Holocene, but this point is hardly a new discovery.

Reply: (1) We thank the reviewer for both positive comments but also some broader critiques that will be helpful to us in clarifying the novelty of this study in our revised submission. Below, we discuss the broad novelty of the paper and how we have clarified that in the revised paper.

First, although many have studied the impact of hydrology on peatland carbon cycles, there is still debate on how that is manifested (i.e. via microbial processes). The conventional viewpoint is that drought can enhance SOC decomposition through the ‘enzyme latch’ mechanism (Fenner and Freeman, 2011). However, neutral or even negative feedbacks between the water level drop and SOC decomposition have been observed (Knorr et al., 2008; Muhr et al., 2011). In a recent paper published in Nature Communications, Wang and colleagues (2017) proposed a new ‘iron gate’ mechanism to interpret the negative relationship between WTD and SOC decomposition in settings with abundant iron. This paper takes a fundamentally different approach from the many excellent but sometimes contradictory modern studies by interrogating past responses. In fact, understanding how the peatland carbon cycle responds to drought on a long term, i.e, centennial or millennial

timescale as discussed here, is even poorer. Our results provide not only a unique historical record but shed new light on a topic of intense contemporary debate. We have added this discussion to the revised manuscript.

Second, the coupled isotopic carbon and hydrogen isotope records reveals evidence for the cumulative (not simply a linear response) effect of drought cycles on isotopic signature and thus on the carbon dynamics. This suggests the enhanced destabilization of peatland carbon stocks if the peatland is subject to the episodic droughts (severe drought and subsequent rewetting). This finding, although admittedly speculative, could be particularly important for peatland management. We have both acknowledged the speculative nature of this conclusion but also further emphasized its potential importance.

Third, we present some important isotopic records that clearly record different drivers than conventional explanations, which might be of significance for future isotopic investigations in peatland. For example, depletion of the carbon isotope composition during dry conditions is opposite to conventional expectations. We also provide an explanation for the influence of different environmental factors on hydrogen isotope composition of different time scale, i.e., source of water vapor for the millennial but RH for the centennial time scales.

(2) The interpretation of isotopic ratios and sources of lipids is reasonable in this study. There is speculation – there always is in the interpretation of carbon and especially hydrogen isotope ratios – but we argue that this speculation is consistent with our data and previous studies. In fact, we have undertaken contextual investigations, such as depth-related changes in water δD values and microbial surveys, that are rather unique for such palaeoclimate investigations and have helped justify our interpretations.

First, the interpretation of lipid source is undoubtedly supported by our previous studies in the Dajiuhu peatland. The sqhC gene data indicates that hopanoids were predominantly synthesized by aerobic bacteria, which is further supported by the close relation between hopanoid concentration and water level in the topmost peat layers (Xie et al., 2013). Consistent with studies in other peatlands, our results clearly suggest that aerial herb plants are the principal contributor for long-chain *n*-alkanes (e.g. C_{29} and C_{31}) preserved in peat samples (Huang et al., 2011, 2014).

Second, we agree that precipitation δD values will affect the leaf wax δD compositions in the peat sequence. However, since there is no to very weak rain amount effect on precipitation δD values in central China (Rao et al., 2016), it is hard to attribute the leaf wax δD values directly to the rainfall amount. Instead, the modern process monitoring (Fig. 2) in this study clearly shows that evaporation, related to RH, modified the vertical and temporal patterns of peat pore water δD values in Dajiuhu. Thus, we use the difference between Dajiuhu leaf wax hydrogen isotope compositions and the nearby Sanbao stalagmite $\delta^{18}\text{O}$ sequence to probe the RH and associated drought events.

Fenner, N., Freeman, C., Drought-induced carbon loss in peatlands. *Nature Geosci.* 4,

895-900 (2011).

- Huang, X., Wang, C., Zhang, J., Wiesenberg, G.L.B., Zhang, Z., Xie, S. Comparison of free lipid compositions between roots and leaves of plants in the Dajihu Peatland, central China. *Geochem. J.* 45, 365-373 (2011).
- Huang, X., Xue, J., Wang, X., Meyers, P.A., Gong, L., Liu, Q., Qin, Y., Wang, H. Hydrologic influence on $\delta^{13}\text{C}$ variations in long-chain *n*-alkanes in the Dajihu peatland, central China. *Org. Geochem.* 69, 114-119 (2014).
- Knorr, K.H., Oosterwoud, M., Blodau, C. Experimental drought alters rates of soil respiration and methanogenesis but not carbon exchange in soil of a temperate fen. *Soil Biol. Biochem.* 40, 1781-1791 (2008).
- Muhr, J., Höhle, J., Otieno, D. O. & Borken, W. Manipulative lowering of the water table during summer does not affect CO₂ emissions and uptake in a fen in Germany. *Ecol. Appl.* 21, 391-401 (2011).
- Rao, Z., Li, Y., Zhang, J., Jia, G., Chen, F. Investigating the long-term palaeoclimatic controls on the δD and $\delta^{18}\text{O}$ of precipitation during the Holocene in the Indian and East Asian monsoonal regions. *Earth-Sci. Rev.* 159, 292-305.
- Wang, Y., Wang, H., He, J.-S., Feng, X. Iron-mediated soil carbon response to water-table decline in an alpine wetland. *Nature Comm.* 8, 15972 (2017).
- Xie, S., Evershed, R.P., Huang, X., Zhu, Z., Pancost, R., Meyers, P.A., Gong, L., Hu, C., Huang, J., Zhang, S., Gu, Y., Zhu, J. Concordant monsoon-driven postglacial hydrological changes in peat and stalagmite records and their impacts on prehistoric cultures in central China. *Geology* 41, 827-830 (2013).

- Organic carbon contents or loss on ignition data should be plotted and they should be measured to very high resolution since these are easy to measure. The data are crucial for supporting authors' argument that mid Holocene period is a time of repeated drought, resulting in excessive burning of peatland carbon and greatly enhanced microbially respired CO₂.

Reply: The TOC data of ZK-5 core were added in the Fig. 4. The TOC profile is consistent with those of our previous ZK-3 core (expressed as particulate organic concentration) and of a nearby core published by Zhang et al. (2016).

It is notable that we did not state that the drought events resulted in “excessive burning” of carbon during the mid-Holocene in Dajihu. What we argued is that the drought event will accelerate peat decomposition and release more microbially respired CO₂. These more nuanced carbon cycle processes related to microbial activities do not have the same impact on TOC content, though TOC content is slightly lower during inferred dry intervals. That is the very reason that we investigate the molecular and isotopic fingerprints of both carbon and hydrogen.

Zhang, W., Yan, H., Cheng, P., Lu, F., Li, M., Dodson, J., Zhou, W., An, Z. Peatland development and climate changes in the Dajihu basin, central China, over the last 14,100 years. *Quat. Int.* 425, 273-281 (2016).

- This paper focuses on Holocene climate and carbon cycling interpretations, but oddly for the past 8 thousand years, there are only 3 radiocarbon dates, whereas there are 14 dates before 8 ka. Large gaps with no radiocarbon dates from 8 to 3.5 ka, and no dates from 3 kyr bp to present. I can see peat accumulation rate is low, but if authors want to claim a new climate record, it is necessary add more radiocarbon dates rather than relying on interpolation alone. There are extremely large changes in sedimentation rate in the past 8 thousand years, with extremely low rate between 8 and 3.5 ka. It is interesting to see the surge in sedimentation rate between ca 3.5 to 3 ka, but then a dramatic slowdown in accumulation from 3 ka to present. These are extremely important results and need to be carefully explained.

Reply: We appreciate that higher resolution dating would add some useful new insights with respect to carbon accumulation rates. Unfortunately, we dated the profile on the basis of the sampling depth resolution, with one dating point for every 15 cm on average (20 for the whole 300 cm profile). However, adding additional data will not change the fundamental observation that peat accumulation rates were low from about 9 to 3.5~ kyr, high from 4-3 kyr and low from 3 kyr to present; we have been able to add one new date that helps refine the duration of the interval of rapid accumulation (i.e., 62 cm below the topmost). Furthermore, the sampling depth shows a strong linear ($R^2 = 0.92$, $n=5$) relationship with the age between 3.4 ka and 8.6 ka, the drought interval discussed here, such that higher resolution dating would appear not to resolve the millennial scale variability in that interval in a way meaningful to our interpretation. Thus, we do not see evidence that a higher resolution age model – which would be difficult to obtain at this stage of the project – would meaningfully change our interpretation.

- Fluxes of n-alkanes and various chain length and distribution data should also be plotted at least in the supplementary data. If hopanoids are analyzed throughout the core with high resolution as shown in figure 1, n-alkane data should also be available. If different hopanoids have different sources as proposed, the fluxes should be plotted separately rather than together as total amounts.

Reply: A new plot including the alkane concentration, CPI, Paq and ACL was added in the revised supplementary material. The drought interval is clearly featured, as expected, by low n-alkane flux, enhanced ACL and low Paq.

It is possible that the taxonomic source of hopanoid producing bacteria will vary over the past 18 ka. However, the meaning of the relationship between hopanoid flux and the water level is based upon the response of whole aerobic bacteria, rather than a specific taxonomic group, which was discussed in our previous publication (Xie et al., 2013).

Xie, S., Evershed, R.P., Huang, X., Zhu, Z., Pancost, R., Meyers, P.A., Gong, L., Hu, C., Huang, J., Zhang, S., Gu, Y., Zhu, J. Concordant monsoon-driven postglacial hydrological changes in peat and stalagmite records and their impacts on prehistoric cultures in central China. *Geology* 41, 827-830 (2013).

- It is hard to understand why the large fluctuation in n-alkane dD values in mid Holocene is not seen in Shabao d18O data. Low relative humidity can increase evaporation of soil waters, which should be seen for cave water isotopic ratios too. Peatland tends to be in places with the most abundant precipitation, it is hard to imagine a larger evaporative isotopic effect in peatland region than where speleothem is formed nearby. Also, during evaporation oxygen isotopic ratios display much greater isotopic effect than hydrogen due to the stronger kinetic isotopic fractionation. A muted oxygen isotope effect along with a large hydrogen isotope change is difficult to explain. If pollen or macrofossil data are available, it is important these data are plotted for comparison. Different plants can have large hydrogen isotopic differences, and it is possible some of the observed hydrogen isotopic fluctuations during the mid Holocene are due to vegetation effects.

Reply:

First, there are many reasons why the peat record might document high resolution hydrological changes not recorded in speleothems. First, much work suggests that bog plant dD values are strongly sensitive to hydrological change, possibly reflecting the sensitivity of wetland vegetation to evapotranspirative processes (Xie et al., 2000; Nichols et al., 2010). Alternatively, this could be due to the sensitivity of shallow water dD values to evaporation – and the fact that peat vegetation incorporates those shallow water signatures, i.e., the unsaturated layers in peat deposits are easily affected by evaporation (Nichols et al., 2010), which is clearly supported by our monitoring of peat pore water dD values (Fig. 2 in the text). Our long-term monitoring on the water level in Dajiuhu also clearly shows that summer drought is common after the ending of the early summer Mei-yu season (and we have added this to the supplementary information).

Fig. S1. Variations of the water level of the peatland (minus values refer water level above the peat surface and vice versa), and the daily rainfall, air relative humidity (RH) in the site of Dajiuhu during 2014-2016.

Second, it is highly debated that speleothem oxygen isotope composition in East Asian monsoon regions could be an indication of precipitation, evaporation, temperature or any other climate signals (Liu et al., 2014; Chen et al., 2016), although the broadly similar pattern found in different sites in east China throughout the whole Holocene could be related to the comparable source water. It is thus hard to evaluate the evaporation effect on the oxygen isotope composition of stalagmite. Related to this, the Sanbao cave is located at a relatively high elevation (1900 m above the sea level), with very thin overlying soil and rock (less than 300 m). Herein the water has a relatively short duration in the uppermost soil, which might escape, to some extent, the influence of evaporation.

Third, we argue that vegetation change is not a main driver on the fluctuations of alkane δD values in the mid-Holocene. C_3 herb plants are the predominant contributor to the long chain n -alkanes preserved in the peat as shown in our preceding studies in Dajiuhu (Huang et al., 2011, 2014), and the vegetation reported before (Zhu et al., 2010) does not show a change comparable with the isotope profiles. For C_3 herb plants, there is a narrow range of $\epsilon_{lipid/water}$ values (Sachse et al., 2012). Thus, the vegetation change could not account for the fluctuations as large as 20–40 per mil during the prominent drought events in the Dajiuhu peat sequence. We have added this to the revised text.

Chen, J., Rao, Z., Liu, J., Huang, W., Feng, S., Dong, G., Hu, Y., Xu, Q., Chen, F. On the timing of the East Asian summer monsoon maximum during the Holocene—Does the speleothem oxygen isotope record reflect monsoon rainfall variability? *Sci. Chin. Earth Sci.* 59, 2328–2338 (2016).

Huang, X., Wang, C., Zhang, J., Wiesenberg, G.L.B., Zhang, Z., Xie, S. Comparison of free lipid compositions between roots and leaves of plants in the Dajiuhu Peatland, central China. *Geochem. J.* 45, 365–373 (2011).

Huang, X., Xue, J., Wang, X., Meyers, P.A., Gong, L., Liu, Q., Qin, Y., Wang, H. Hydrologic influence on $\delta^{13}C$ variations in long-chain n -alkanes in the Dajiuhu peatland, central China. *Org. Geochem.* 69, 114–119 (2014).

Liu, Z., Wen, X., Brady, E. C., Otto-Bliesner, B., Yu, G., Lu, H., Cheng, H., Wang, Y., Zheng, W., Ding, Y., Edwards, R. L., Cheng, J., Liu, W., Yang, H. Chinese cave records and the East Asia summer monsoon. *Quat Sci Rev* 83, 115–128 (2014).

Nichols, J., Booth, R.K., Jackson, S.T., Pendall, E.G., Huang, Y. Differential hydrogen isotopic ratios of *Sphagnum* and vascular plant biomarkers in ombrotrophic peatlands as a quantitative proxy for precipitation–evaporation balance. *Geochim. Cosmochim. Acta* 74, 1407–1416 (2010).

Sachse, D., Billault, I., Bowen, G. J., Chikaraishi, Y., Dawson, T. E., Feakins, S. J., Freeman, K. H., Magill, C. R., McInerney, F. A., van Der Meer, M. T. J., Polissar, P., Robins, R. J., Sachs, J. P., Schmidt, H.-L., Sessions, A. L., White, J. W. C., West, J. B., Kahmen A. Molecular paleohydrology: interpreting the hydrogen-isotopic composition of lipid biomarkers from photosynthesizing organisms. *Annu. Rev.*

Earth Planet. Sci. 40, 221–249 (2012).

Xie, S., Nott, C. J., Avsejs, L. A., Volders, F., Maddy, D., Chambers, F. M., Gledhill, A., Carter, J. F., Evershed R. P., Palaeoclimate records in compound-specific dD values of a lipid biomarker in ombrotrophic peat. *Org. Geochem.* 31, 1053–1057 (2000).

Zhou, W., Yu, X., Jull, A.J.T., Burr, G., Xiao, J., Lu, X., Xian, F. High-resolution evidence from southern China of an early Holocene optimum and a mid-Holocene dry event during the past 18000 years. *Quat. Res.* 62, 39–48 (2004).

- I actually do not see clear correspondence of low n-alkane dD values with enhanced hopanoid fluxes. There are only a few points with relatively high hopanoid fluxes but these do not directly correspond to high n-alkane dD values (indicating drought as argued). I also fail to see how n-alkane high dD values are correlated to hopanoid d13C changes. If authors want to show they are correlated show how, they should plot X-Y plots for the same samples to see if there are any real correspondence.

Reply: The correlation coefficient (r) between n -C29 δD values and the hopanoid flux is 0.26 ($p=0.05$). However, there is a more fundamental point here that we have tried to make clear in the paper – hopanoid and n -alkane signals will not be directly correlated in a given horizon due to stratigraphic offsets arising from depth of production, i.e. during a dry and oxidizing interval, enhanced hopanoid production could be occurring in the peat subsurface. Consequently, the main text emphasizes the broad time interval of variation in all of these parameters. However, if we bundle intervals at 1-3ka, 3-5 ka, 5-7.2ka, 7.2-9.2 ka, 9.2-11.6 ka and 11.6-12.8 ka, then we observe a relatively strong correlation ($r=0.9$; Fig. S2), largely based on data clustering into low hopanoid/low dD and high hopanoid/high dD groups and supporting our interpretation. In particular, both proxies indicate prolonged drought at 11.6-10.6 ka, and 7-3 ka. A supplementary figure was added in the Supplementary information.

Fig. S2. Correlation plot between n -C29 δD values and the hopanoid flux.

- Interpreting different hopanoids have different sources based on d13C data alone is quite a stretch. Often there are natural differences in isotopic ratios between different hopanoids, even if they are made by the same organism. Note some hopanoids can be produced by fern as well.

Reply: It is true that different hopanoids will have quite different $\delta^{13}\text{C}$ values. That is why we focus on the $\delta^{13}\text{C}$ variations of a single compound (e.g. C_{29} $\beta\beta$ hopane) in the discussion. For that hopanoid, the variations observed here are much larger than simple natural differences within a given organism – hopanoid $\delta^{13}\text{C}$ values are almost always above -35 per mil in peats and soils and frequently much higher as noted in the citations. In fact, in the investigations of hopanoids extracted from Sphagnum growing in modern peatlands, hopanoids with $\delta^{13}\text{C}$ values between -34 and -40 per mil have been attributed to the input from methanotrophic bacteria (Raghorbarsing et al., 2005; van Winden et al., 2010). As such, the very depleted $\delta^{13}\text{C}$ values of individual hopanoids almost certainly indicate a greater contribution from methanotrophic organisms.

It has been reported that ferns can produce some hopene in very low concentration (Ageta and Arai, 1983), but this is a different hopanoid than those discussed here. Furthermore, in peatlands like Dajiuhu, ferns always grow on the relatively dry land, and are not the main peat-forming plants (Luo et al., 2015). Most importantly, the Ageta and Arai finding has never been duplicated.

Ageta, H., Arai, Y. Fern constituents: pentacyclic triterpenoids isolated from *Polypodium niponicum* and *P. formosanum*. *Phytochemistry* 22, 1801–1808 (1983).

Luo, T., Lun, Z., Gu, Y., Qin, Y., Zhang, Z., Zhang, B. Plant community survey and ecological protection of Dajiuhu wetlands in Shennongjia area. *Wetland Science* 13(2), 153-160 (2015).

Raghoebarsing, A.A., Smolders, A.J.P., Schmid, M.C., Rijpstra, W.I.C., Wolters-Arts, M., Derksen, J., Jetten, M.S.M., Schouten, S., Sinninghe Damsté, J.S., Lamers, L.P.M., Roelofs, J.G.M., Op den Camp, H.J.M., Strous, M. Methanotrophic symbionts provide carbon for photosynthesis in peat bogs. *Nature* 436, 1153-1156 (2005).

van Winden, J. F., Kip, N., Reichart, G. J., Jetten, M. S. M., Op den Camp, H. J. M., Sinninghe Damsté, J. S. Lipids of symbiotic methane-oxidizing bacteria in peat moss studied using stable carbon isotopic labelling. *Org. Geochem.* 41, 1040-1044 (2010).

- There is some correlation between dD and d13C for C29 n-alkane for mid Holocene drought section but not particularly strong. To say low d13C values during the proposed dry phases originates from ambient CO2 with greater proportion of recycled CO2 is not convincing to me. If the peat land has stopped accumulating as suggested by extremely low accumulation rate during this time, there would not be so much carbon to burn to produced CO2 with low d13C values. Again there is no TOC data plotted in this paper hence difficult to judge for me. Often it is during the wet times, increased methane production and resulting microbially respired CO2 from methane with low d13C values could contribute low d13C values of CO2, rather than the opposite. Also, as discussed in the paper, dryer conditions can lead to reduced carbon isotopic fractionation of higher plants, and higher d13C values which can then counter the proposed source carbon effect. It is complicated.

Reply: First of all, we agree that the controls on leaf wax $\delta^{13}\text{C}$ values are complicated. In fact, we explicitly emphasize that our observations are unexpected based on assumptions of plant and peatland carbon cycle responses to a drier climate. This is the very foundation for our interpretation that more nuanced carbon cycle responses must be occurring. We provide a more detailed response to the reviewer's points below:

(1) The relatively weak correlation between δD and $\delta^{13}\text{C}$ of C_{29} n-alkane results from the non-linear response of carbon biogeochemical process to drought conditions related to the deuterium content. We state that more clearly in the revised manuscript.

(2) During the drought intervals in the mid-Holocene, the quite low accumulation rate (rather than low TOC content) could suggest that the bulk peat deposit has been decomposed to release as respired CO_2 , which can be used for plant synthesis and result in quite low $\delta^{13}\text{C}$ values of n-alkanes. In addition, although the methane production could not increase in drought intervals, the aerated peat surface makes methane easily oxidize to CO_2 , which could be another source of ^{13}C -depleted CO_2 in the topmost peat layers (see Zheng et al., 2014).

(3) The influence of drought on leaf carbon isotope fractionation indeed can counter the proposed source carbon effect. But that would mean that there was an even strong respiration signal that has been partially masked. We do not say this explicitly as we do not want to overly speculate – but it is implicit in our text given our framing around the ‘unexpected’ $\delta^{13}\text{C}$ response to drought.

- Figure 2 needs some dashed lines to separate various depth sections. The number of water samples for each collection season needs to be stated clearly regardless whether the data have been published before. Are there daily water sample collections or weekly collections for example?

Reply: Figure 2 has been modified on the basis of the reviewer's comment, by adding the dashed lines in the figure and sample numbers in the figure caption. The water sample was only collected in a single day during each field expedition and this has been clarified in the methods.

- If pronounced dry episodes in mid Holocene are already observed in China, authors should plot these data in this paper to help readers.

Reply: We are the first to document the dry episodes in mid-Holocene in middle Yangtze in 2013 by measuring the hopanoid flux in ZK03 of the Dajiuhu peatland (Xie et al., 2013), and this was plotted in Figure 4. The mid-Holocene drought is now supported by the magnetic parameters measured in the stalagmite of the neighboring Heshang cave (Zhu et al., 2017). The dry episodes in mid-Holocene was not only identified in central China but also in northwest China recently (Rao et al., 2016). For more detailed information on mid-Holocene drought, please refer the synthesis in Rao et al. (2016).

Xie, S., Evershed, R.P., Huang, X., Zhu, Z., Pancost, R., Meyers, P.A., Gong, L., Hu, C., Huang, J., Zhang, S., Gu, Y., Zhu, J. Concordant monsoon-driven postglacial hydrological changes in peat and stalagmite records and their impacts on prehistoric cultures in central China. *Geology* 41, 827-830 (2013).

Zhu, Z., Feinberg, J. M., Xie, S., Bourne, M., Huang, J., Hu, C., Cheng, H. Holocene ENSO-related cyclic storms recorded by magnetic minerals in speleothems of central China. *P. Natl. Acad. Sci. USA*, 114, 852-857(2017).

Rao, Z., Li, Y., Zhang, J., Jia, G., Chen, F. Investigating the long-term palaeoclimatic controls on the δD and $\delta^{18}O$ of precipitation during the Holocene in the Indian and East Asian monsoonal regions. *Earth-Sci. Rev.* 159, 292-305.

Response to comments from Reviewer #3:

This manuscript presents $\delta^{2}H_{wax}$ and $\delta^{13}C$ data from a peatbog in northern China to identify drought events. The scientific methods and data seem sound and the $\delta^{13}C$ data is interesting. However, there are a few items missing from the manuscript. First, I think this is an important data set, but the authors could bring out the wider implications of it more strongly.

In general, I think the discussion regarding the $\delta^{2}H$ record needs to be clearer. The diversion between Sanbao and Dajiuhu from 8 and 5 ka (broadly speaking) suggesting drought conditions does seem reasonable and important. However, I find the discussion regarding the match with the Bay of Bengal to be confusing and potentially erroneous. Regarding lines 132-137 in particular- are the authors suggesting that vapor source (and thereby wind direction) is a control on $\delta^{2}H$ values? It is certainly surprising that the $\delta^{2}H$ values at Dajiuhu would be similar to the Bay of Bengal $\delta^{2}H$ values given that Dajiuhu is dominated by the EASM and the Bay of Bengal reflects the ISM. While this is not to say that the EASM and ISM cannot be synchronous, increasingly it is becoming clear that they appear to behave asynchronously in the Holocene. This leads me to related point, which is that the authors don't make clear which monsoon system they are referring to.

Reply: According to the modern investigation of moisture transport in Asia, air mass from the Indian Ocean contributes the most vapor to the most part of the eastern China (see below Fig. S3), including Dajiuhu peatland studied here (Fig. S3; Ding et al., 2004). Thus, it is not odd to observe a similar general trend between the Dajiuhu $\delta^{2}H$ sequence and that from the Bay of Bengal. We have added a discussion of this point to the revised manuscript, however, it should be noted this is not the main focus of the paper.

Fig. S3. Moisture transport patterns (averaged for 1990-1999) after the onset (the 5th pentad of May–the 2nd pentad of July) of the South China Sea summer monsoon (Ding et al., 2004).

The location of Dajiuhu was labelled as a red circle.

Ding, Y., Li, C., Liu, Y. Overview of the South China Sea Monsoon experiment. *Adv. Atmos. Sci.* 21, 343–360 (2004).

Lines 66-68- Bowen's model has been shown to have some deficiencies, particularly with regard to higher altitude locations. I would be more comfortable with this interpretation if actual precipitation samples were used to demonstrate this.

Reply: We agree that Bowen's model may have deficiency when applying to a specific site. In the revised supplementary material, we added a new table to show the current available data on the precipitation δD values in Dajiuhu. These actual rainfall δD data are less negative than the one calculated from Bowen's model. Regardless, the difference between the measured and modeled precipitation δD values does not affect the conclusion that evaporation affects the pore water δD values and the subsequent leaf wax δD features preserved in the peat sequences. But we agree with the reviewer that a more rigorous treatment of this will be useful to future readers. We have revised the text according this comment.

Lines 125-160- This is more of a discussion rather than a presentation of results. Move accordingly.

Reply: Have moved to the discussion part as suggested.

Lines 130-131- What about the possibility that vegetation in this peatland may have been different, particularly during the glacial period?

Reply: According to the pollen data published in Dajiuhu, Cyperaceae and *Sphagnum* were the dominant wetland species (Shi et al., 2008; Zhu et al., 2010). *Carex* and *Sphagnum* are

also the major peat-forming plants in contemporary Dajiuhu and many other peatlands in China. In this way, herb plants are the predominant source of C₂₉ and C₃₁ *n*-alkanes over the last 18 ka in this study.

Zhu, C., Ma, C., Yu, S.-Y., Tang, L., Zhang, W., Lu, X. A detailed pollen record of vegetation and climate changes in Central China during the past 16000 years. *Boreas* 39, 69-76 (2010).

Shi, M., Yu, J., Gu, Y., Chen, J. Climate changes in interim of late Pleistocene and Holocene in Dajiuhu Basin of Shennongjia, Hubei Province: evidence from Pollen. *Geol. Sci. Tech. Infor.* 27(6), 24-28 (2008).

Lines 138-140- Identify which particular periods you are referring to where there are prominent differences. I see some in Figure 4, but it's not clear if we are looking at the same time frame.

Reply: Revised.

Lines 154-156- Indeed, middle Holocene drought conditions are a prevalent feature of many ISM records. There are fewer EASM records that show this though and I think this point needs expansion.

Reply: The air water mass of the Dajiuhu peatland is mainly contributed by ISM but might be influenced, to some extent, by the East Asian summer monsoon. This is further shown in modern air water mass (Fig. S3, see above). It is thus not surprising that the mid-Holocene dry condition occurred at the Dajiuhu peatland, a phenomenon of ISM as stated by the reviewer. As mentioned above, we have added a discussion of this issue to the revised text.

Figure 4- Some annotation on the $\Delta\delta^2\text{H}$ panel would be helpful, indicating which axis direction indicates what.

Reply: Added.

Reviewers' comments:

Reviewer #1 (Remarks to the Author):

I repeat my summary of this contribution from my previous review of it:

"Huang and his colleagues interweave a suite of organic geochemical paleoenvironmental proxies in this elegant and interesting contribution. Their goal is to explore for evidence that some of the abundant amounts of organic carbon sequestered in peatlands can be released and recycled and to identify the situations when this process might occur. To this end, they combine compound-specific carbon and hydrogen isotope measurements of leaf-wax components that represent the principal plants responsible for peat accumulation with compound-specific carbon isotope measurements of biomarkers for aerobic bacteria that would be responsible for much of the carbon recycling. The hydrogen isotope measurements are indicators of former wet-dry alternations in the Dajiuhe peatland that the authors study. These proxies are complemented by oxygen isotope variations recorded in the nearby Shanbao Cave to allow the authors to reconstruct the mid-Holocene paleohydrologic history of this area. What they conclude is that during episodes of drier climate and depressed water levels, peat was subject to bacterial oxidation that released isotopically light carbon dioxide that was assimilated by the peat-forming plants, yielding more negative carbon isotopic values of their leaf waxes. This chain of events makes biogeochemical sense, yet it has not been so elegantly documented before."

The authors have extensively revised their contribution, making good use of the thoughtful and thought-provoking comments provided by the two other reviewers. The contribution is much improved. Of particular significance to me, its readability has improved. I am satisfied with the revisions made by the authors. I think the contribution merits publication in Nature Communications.

Phil Meyers, September 8, 2017

Reviewer #2 (Remarks to the Author):

I read the revised version of the paper more carefully. Authors addressed some of my points, but there are a number of important points not being addressed sufficiently. I have also discovered more problems with the paper as I read the paper again. Here are a number of major problems I found:

- Authors must tabulate all measured isotope data. It is unacceptable to not tabulate all the measured isotope data (with standard deviation) for all compounds for close examination by readers – that is what supplementary information is for. It is particularly important for this paper because I have found several places where the stated data manipulation in the text does not match the figures. The biggest discrepancy I see is the data points plotted in Fig.4 a and Fig.4 f. Since Fig.4 f is derived from the difference between the Shanbao cave carbonate isotope data and Fig.4 a (measured hydrogen isotope data) as stated in the figure legend, there should be an equal number of data points. However, there are more points plotted in Fig.4f than Fig.4 a, especially for the critical interval where authors claim "mid Holocene droughts". It almost looks as if the authors invented new data points by doing linear interpolation of their data between points to add a false impression of higher sampling resolution. For example, there are only 4 data points between 5.5 and 4.4 ka in Fig.4a, but after subtraction to cave isotope data, mysteriously at least 7 points are plotted in Fig.4f. Same is true for other claimed mid-Holocene δD excursions.

- Actually I do not believe any of the so-called mid-Holocene drought events – the authors claim they are the first to report these. I requested authors adding more radiocarbon dates for the mid Holocene but the request is not honored. Ok, but authors cannot claim big droughts with only 3 to

4 points of very low resolution hydrogen isotope values in Fig.4a (also Fig.4f appear to artificially added more points by linear interpolation). If the drought interval really is that interesting, it is very important for authors to add more data points (i.e., higher sampling resolution) to see these few unusually hydrogen isotopic points indeed represent systematic changes.

- Age model plot in the supplementary info. The stated faster and slower peat accumulation rates are opposite to data: obviously accumulation rate is lowest from 3 to 8 kyr.

-

- Fig.3. vertical δD plot is way out of scale, the -198 and -204 have 6 per mil differences, especially considering errors. These are statistically the same. The plot has nothing new but adds confusion. Why would peat n-alkanes have 6 per mil lower hydrogen isotopic value than plant n-alkanes? If you measure another batch of plants and peat samples, can you reproduce this minor difference? Surely the answer is no. Making a figure with out of scale Y axis illustrate virtually nothing.

- The repeated claimed close match between n-alkane δD and $\delta^{13}C$ in supplementary Fig.4 is not true. δD in mid Holocene has a few big excursion points, but corresponding $\delta^{13}C$ do not show any out of ordinary excursions. In the contrary, a few of large carbon isotopic excursion to lower values is not accompanied by corresponding hydrogen isotopic increases. This point that higher hydrogen isotopic excursions (proposed to indicate droughts) is accompanied by more respired carbon dioxide being assimilated by plants is therefore false. In the abstract the paper even claims synchronous hydrogen and carbon isotope change – I do not see that in the data. Line 218-219: the statement “Each of the three Holocene positive δ^2H shifts is associated with a decrease in leaf wax $\delta^{13}C$ values” is completely not true. Many H and C isotope changes do not match at all.

- If authors want to make the claim for droughts in mid Holocene, they should measure hydrogen isotopic values of leaf wax n-alkanes at much higher resolution. Why not measure these at the same resolution as hopanoid concentrations are measured in Fig.4c? hopanoids have much lower concentrations than n-alkanes yet their isotopic values are measured at higher resolution. This does not make sense to me.

- What is the physical mechanism for pronounced mid Holocene drought in the region? I requested in my previous review that authors plot the supporting data, but the request is not honored. It is not my responsibility to read other papers that authors of this paper claim support conclusions of this paper. Making such claim of big mid Holocene droughts requires extensive evidence – I feel the paper seems to treat this serious claim in a very casual manner. It is really dangerous to make such serious claim based on a few very low resolution odd data points. It is very important to plot other support data in this paper, and examine the spatial extent of droughts, and most importantly of all, propose some reasonable physical mechanisms to explain the origin of these major climate variations.

- TOC plot should show the number of samples analyzed and all data must all be tabulated for close evaluation. All concentrations of compounds must be provided as tables, not just plots.

- What is carbon accumulate rate during the mid Holocene? Use TOC and age model to plot carbon accumulation rate.

- I do not believe higher amounts of hopanoids are necessarily a result of enhanced methane oxidation. Bacterial production of hopanoids can be affected by all kinds of environmental and biological factors. Bacterial populations change under different climate and environmental conditions, and production rate of hopanoids differ in different populations.

- Line 178: to say Sanbao has a relatively high sea level of 1900 m asl but do not mention altitude of DaJuhu in the paper is odd. Dajuhu is 1760 m asl. Not much difference. There is no evidence to say 140 m difference in elevation will make Sanbao site water escape evaporation but Dajuhu does

not. In all soil profiles, if sampled against depth, you will see the same isotope trend. Fig.2 and test seems to say such isotope profile occurs in Dajuhu but not in Sanbao – what is the evidence?

Reviewer #3 (Remarks to the Author):

The authors have satisfactorily addressed my comments. I think the manuscript is much stronger with the wider implications written in very clearly.

Response to comments from Reviewer #1:

I repeat my summary of this contribution from my previous review of it:

"Huang and his colleagues interweave a suite of organic geochemical paleoenvironmental proxies in this elegant and interesting contribution. Their goal is to explore for evidence that some of the abundant amounts of organic carbon sequestered in peatlands can be released and recycled and to identify the situations when this process might occur. To this end, they combine compound-specific carbon and hydrogen isotope measurements of leaf-wax components that represent the principal plants responsible for peat accumulation with compound-specific carbon isotope measurements of biomarkers for aerobic bacteria that would be responsible for much of the carbon recycling. The hydrogen isotope measurements are indicators of former wet-dry alternations in the Dajiuhu peatland that the authors study. These proxies are complemented by oxygen isotope variations recorded in the nearby Shanbao Cave to allow the authors to reconstruct the mid-Holocene paleohydrologic history of this area. What they conclude is that during episodes of drier climate and depressed water levels, peat was subject to bacterial oxidation that released isotopically light carbon dioxide that was assimilated by the peat-forming plants, yielding more negative carbon isotopic values of their leaf waxes. This chain of events makes biogeochemical sense, yet it has not been so elegantly documented before."

The authors have extensively revised their contribution, making good use of the thoughtful and thought-provoking comments provided by the two other reviewers. The contribution is much improved. Of particular significance to me, its readability has improved. I am satisfied with the revisions made by the authors. I think the contribution merits publication in Nature Communications.

Reply: We are grateful for the positive comments.

Response to comments from Reviewer #2:

I read the revised version of the paper more carefully. Authors addressed some of my points, but there are a number of important points not being addressed sufficiently. I have also discovered more problems with the paper as I read the paper again. Here are a number of major problems I found:

- Authors must tabulate all measured isotope data. It is unacceptable to not tabulate all the measured isotope data (with standard deviation) for all compounds for close examination by readers – that is what supplementary information is for. It is particularly important for this paper because I have found several places where the stated data manipulation in the text does not match the figures. The biggest discrepancy I see is if the data points plotted in Fig.4 a and Fig.4 f. Since Fig.4 f is derived from difference between the Shanbao cave carbonate isotope data and Fig.4 a (measured hydrogen isotope data) as stated in the figure legend, there should be equal number of

data points. However, there are more points plotted in Fig.4f than Fig.4 a, especially for the critical interval where authors claim “mid Holocene droughts”. It almost looks as if the authors invented new data points by doing linear interpolation of their data between points to add false impression of higher sampling resolution. For example, there are only 4 data points between 5.5 and 4.4 ka in Fig.4a, but after subtraction to cave isotope data, mysteriously at least 7 points are plotted in Fig.4f. Same is true for other claimed mid-Holocene δD excursions.

Reply: We have added the isotopic data to the Supplementary information as suggested.

Yes, the stalagmite record in Fig. 4f has a much higher resolution than our peatland record of ZK-5 in Fig. 4a, so we conducted a linear interpolation for the peatland record before calculating the difference between the two records in Fig. 4e. Since this seems to have caused a misunderstanding, in the revised text, we discard the linear interpolation and utilized the data with comparable ages (mostly with an age difference of <0.02 ka) from the two data sets to calculate the difference.

- Actually I do not believe any of the so-claimed mid-Holocene drought events – the authors claim they are the first to report these. I requested authors adding more radiocarbon dates for the mid Holocene but the request is not honored. Ok, but authors cannot claim big droughts with only 3 to 4 points of very low resolution hydrogen isotope values in Fig.4a (also Fig.4f appear to artificially added more points by linear interpolation). If the drought interval really is that interesting, it is very important for authors to add more data points (i.e., higher sampling resolution) to see these few unusually hydrogen isotopic points indeed represent systematic changes.

Reply: Firstly, we did ‘honor’ the request to add more radiocarbon dates; it is a valid suggestion; however, it is not possible at this time and our argument was that although it would be useful, it was not necessary for this specific paper.

We entirely agree that it is an excellent suggestion to improve the argument for drought in the mid-Holocene. It seems that the critical issue to the reviewer is that we are arguing based on limited data for a new climatic event in central China; we entirely understand why that would be inappropriate and require more evidence. However, several papers in recent years have discussed the increase in aridity during the mid-Holocene in central China on the basis of sedimentary records as well as climate models (Xie et al., 2013; Dallmeyer et al., 2013; Jiang et al., 2013; Zheng et al., 2013; Zhu et al., 2017); therefore, we have added a paragraph to emphasize the status of these discussions, though we stress that the discussion of the drying mid-Holocene is not the main point of our current paper. We also note that perhaps the term drought – although used by others – is not justified by our data which instead show only evidence for relatively dryer conditions in the peat. Therefore, we have clarified the language, which hopefully ameliorates the Reviewer’s concerns.

The other suggestion was to increase the resolution for the mid-Holocene drought interval. Again, we agree that a higher resolution record would be useful. However, the current data were based on analysis of samples from 1-cm interval for the mid-Holocene dry interval,

which is quite high depth resolution, actually typical, for peat sequences. Such a low peat accumulation is consistent with extended dry conditions, which are entirely consistent with organic matter degradation.

(NOTE: The difference in resolution between δD and hopane data is discussed below)

Dallmeyer, A., Claussen, M., Wang, Y., Herzschuh, U. Spatial variability of Holocene changes in the annual precipitation pattern: a model-data synthesis for the Asian monsoon region. *Clim. Dyn.* 40, 2919-2936 (2013).

Jiang, D., Tian, Z., Lang, X. Mid-Holocene net precipitation changes over China: model-data comparison. *Quaternary Sci. Rev.* 82, 104-120 (2013).

Xie, S., Evershed, R.P., Huang, X., Zhu, Z., Pancost, R., Meyers, P.A., Gong, L., Hu, C., Huang, J., Zhang, S., Gu, Y., Zhu, J. Concordant monsoon-driven postglacial hydrological changes in peat and stalagmite records and their impacts on prehistoric cultures in central China. *Geology* 41, 827-830 (2013).

Zheng, W., Wu, B., He, J., Yu, Y. The East Asian summer monsoon at mid-Holocene: results from PMIP3 simulations. *Clim. Past* 9, 453-466 (2013).

Zhu, Z., Feinberg, J.M., Xie, S., Bourne, M.D., Huang, C., Hu, C., Cheng, H. Holocene ENSO-related cyclic storms recorded by magnetic minerals in speleothem of central China. *P. Natl. Acad. Sci. USA* 114, 852-857 (2017).

- Age model plot in the supplementary info. The stated faster and slower peat accumulation rates are opposite to data: obviously accumulation rate is lowest from 3 to 8 kyr.

Reply: We acknowledge this error and it has been corrected.

- Fig.3. vertical δD plot is way out of scale, the -198 and -204 have 6 per mil differences, especially considering errors. These are statistically the same. The plot has nothing new but adds confusion. Why would peat n-alkanes have 6 per mil lower hydrogen isotopic value than plant n-alkanes? If you measure another batch of plants and peat samples, can you reproduce this minor difference? Surely the answer is no. Making a figure with out of scale Y axis illustrate virtually nothing.

Reply: Thank you for noticing this – the figure has been deleted in the revised text.

- The repeated claimed close match between n-alkane δD and $\delta^{13}C$ in supplementary Fig.4 is not true. δD in mid Holocene has a few big excursion points, but corresponding $\delta^{13}C$ do not show any out of ordinary excursions. In the contrary, a few of large carbon isotopic excursion to lower values is not accompanied by corresponding hydrogen isotopic increases. This point that higher hydrogen isotopic excursions (proposed to indicate droughts) is accompanied by more respired carbon dioxide being assimilated by plants is therefore false. In the abstract the paper even claims synchronous hydrogen and carbon isotope change – I do not see that in the data. Line 218-219: the statement “Each of the three Holocene positive δ^2H shifts is associated with a decrease in leaf wax $\delta^{13}C$ values” is completely not true. Many H and C isotope changes do not match at all.

Reply: In the revised Fig. 4, vertical dashed lines were added to help identify the close relationship between n-alkane δD and $\delta^{13}C$. As shown in this new Fig. 4, each of the positive

$\delta^2\text{H}$ shifts is closely, though non-linearly, associated with a decrease in *n*-alkane $\delta^{13}\text{C}$.

- If authors want to make the claim for droughts in mid Holocene, they should measure hydrogen isotopic values of leaf wax *n*-alkanes at much higher resolution. Why not measure these at the same resolution as hopanoid concentrations are measured in Fig.4c? hopanoids have much lower concentrations than *n*-alkanes yet their isotopic values are measured at higher resolution. This does not make sense to me.

Reply: Unfortunately, hopanoid concentrations and *n*-alkanes isotope compositions were determined in different peat cores of the peatland. Measurement of the hopanoid flux data was derived from a neighboring peat core (ZK-3, retrieved in 2005), and leaf wax δD values were determined for the new ZK-5 core (retrieved in 2013). These two cores have different accumulation rates during the mid-Holocene, possibly resulting from their specific location in the basin. Unfortunately, the ZK-3 core has now been exhausted by extensive samplings for a variety of investigations, thereby precluding further analyses. Furthermore, due to the addition of perdeuterated standard, the fractions generated for the ZK-3 biomarker investigation cannot be used for compound-specific δD analyses. In fact, it was largely for these reasons that we obtained the new core ZK-5.

- What is the physical mechanism for pronounced mid Holocene drought in the region? I requested in my previous review that authors plot the supporting data, but the request is not honored. It is not my responsibility to read other papers that authors of this paper claim support conclusions of this paper. Making such claim of big mid Holocene droughts requires extensive evidence – I feel the paper seems to treat this serious claim in a very casual manner. It is really dangerous to make such serious claim based on a few very low resolution odd data points. It is very important to plot other support data in this paper, and examine the spatial extent of droughts, and most importantly of all, propose some reasonable physical mechanisms to explain the origin of these major climate variations.

Reply: The drier conditions in the mid-Holocene are not the main point of our paper. This paper aims to explore the relationship between drier conditions and peatland carbon cycling on centennial timescales by combining leaf wax carbon and hydrogen isotopic compositions. However, we have added an extensively referenced paragraph discussing the mid-Holocene drying in the middle reaches of Yangtze River. Based on our previous work, data in this study and the extensive work of other researchers, we are confident that drier conditions prevail in the middle and lower reaches of the Yangtze River during the mid-Holocene (Xie et al., 2013; Dallmeyer et al., 2013; Jiang et al., 2013; Zheng et al., 2013; Zhu et al., 2017). The drier condition during the mid-Holocene in central China contrasts with the proposed wet interval of 8-3 ky in the north China and south China (Rao et al., 2016).

As for the mechanism, given that this is not the point of the paper, we are cautious about over-explanation. However, a previous study interpreted such a spatial pattern as the influence of the Western Pacific Subtropical High (WPSH) and the associated ENSO variance. During the mid-Holocene, the west-east surface sea temperature gradient is strong (Koutavous and Joanides, 2012), and thus the average position of WPSH will move north and west, making the middle and lower reaches of the Yangtze River dominate by downdraft.

In fact, a similar mechanism has been proposed to interpret the negative relationship between precipitation in the middle and lower reaches of the Yangtze River and the summer monsoon intensity on decade timescale (Ding et al., 2008).

Ding, Y.H., Wang, Z.Y., Sun, Y. Inter-decadal variation of the summer precipitation in East China and its association with decreasing Asian summer monsoon. Part I: observed evidences. *Int. J. Climatol.* 28, 1139-1161 (2008).

Koutavas, A., Joanides, S. El Niño–Southern Oscillation extrema in the Holocene and Last Glacial Maximum. *Paleoceanography* 27, PA4208 (2012).

Rao, Z., Li, Y., Zhang, J., Jia, G., Chen, F. Investigating the long-term palaeoclimatic controls on the δD and $\delta^{18}O$ of precipitation during the Holocene in the Indian and East Asian monsoonal regions. *Earth-Sci. Rev.* 159, 292-305 (2016).

- TOC plot should show the number of samples analyzed and all data must all be tabulated for close evaluation. All concentrations of compounds must be provided as tables, not just plots.

Reply: We have added these data to the Supplementary file.

- What is carbon accumulate rate during the mid Holocene? Use TOC and age model to plot carbon accumulation rate.

Reply: The TOC content in peatlands is general very high and is relatively constant during most time in the Holocene, and thus the carbon accumulation rate is mainly related to the deposition rate (Supplementary Fig. 1 and Table 1), i.e. peat accumulation rate is relatively slower during 3 to 8 kyr (dry period) and faster during 2-3, 9-10 and 16-18 kyr.

- I do not believe higher amounts of hopanoids are necessarily a result of enhanced methane oxidation. Bacterial production of hopanoids can be affected by all kinds of environmental and biological factors. Bacterial populations change under different climate and environmental conditions, and production rate of hopanoids differ in different populations.

Reply: We agree and we did not attribute the higher hopanoid accumulation rate to enhanced methane oxidation. Actually, our previous studies of *sqhC* functional genes (Xie et al., 2013; Gong et al., 2015) provided strong evidence that the hopanoids in Dajiuhu were predominantly biosynthesized by aerobic bacteria other than methanotrophs. In addition, the modern process investigation clearly showed that the geological hopanoids have a close relationship with the water table in the Dajiuhu peatland (Xie et al., 2013). We do argue that strong depletion in the carbon isotopic composition of hopanoids could be evidence that changes in methane cycling were part of an impacted carbon cycle during periods of drought. he observed changes are entirely consistent with those observed in previous experiments and indicate only a subtle shift in the source of hopanoids that imprints this isotopic signature (van Winden et al., 2010, 2012).

Gong, L., Wang, H., Xiang, X., Qiu, X., Liu, Q., Wang, R., Zhao, R., Wang, C. pH shaping the composition of *sqhC*-containing bacterial communities. *Geomicrobiol. J.* 32, 433-444 (2015).

Xie, S., Evershed, R.P., Huang, X., Zhu, Z., Pancost, R., Meyers, P.A., Gong, L., Hu, C.,

- Huang, J., Zhang, S., Gu, Y., Zhu, J. Concordant monsoon-driven postglacial hydrological changes in peat and stalagmite records and their impacts on prehistoric cultures in central China. *Geology* 41, 827-830 (2013).
- van Winden, J. F., Talbot, H. M., Kip, N., Reichart, G. J., Pol, A., McNamara, N. P., Jetten, M. S. M., Op den Camp, H. J. M., Sinninghe Damsté, J. S. Bacteriohopanepolyol signatures as markers for methanotrophic bacteria in peat moss. *Geochim. Cosmochim. Acta* 77, 52-61(2012).
- van Winden, J. F., Kip, N., Reichart, G. J., Jetten, M. S. M., Op den Camp, H. J. M., Sinninghe Damsté, J. S. Lipids of symbiotic methane-oxidizing bacteria in peat moss studied using stable carbon isotopic labelling. *Org. Geochem.* 41, 1040-1044 (2010).

- Line 178: to say Sanbao has a relatively high sea level of 1900 m asl but do not mention altitude of DaJuhu in the paper is odd. Dajuhu is 1760 m asl. Not much difference. There is no evidence to say 140 m difference in elevation will make Sanbao site water escape evaporation but Dajuhu does not. In all soil profiles, if sampled against depth, you will see the same isotope trend. Fig.2 and test seems to say such isotope profile occurs in Dajuhu but not in Sanbao – what is the evidence?

Reply: It is true that the two sites have a small difference in altitude. However, the evaporation could be impacted by the vegetation cover in addition to the elevation difference. Sanbao site is located in the Shennongjia nature reserve featured by the dense forest canopy and deeper soil layers (normally >2m), from which the drip water of the Sanbao cave comes. In contrast, the Dajuhu peatland is mainly covered with herb plants, which use soil water from the uppermost 20-30cm, thereby recording a strong evaporation effect within their biochemicals.

Another source evidence that indicates evaporation shows little effect on stalagmite $\delta^{18}\text{O}$ in Sanbao site is that Holocene stalagmite $\delta^{18}\text{O}$ sequences show high consistency from southwest China to northeast China, spanning a distance >2000 km across quite different climate zones (Liu et al., 2015; Chen et al., 2016). In this case, evaporation only has very minor effect on the stalagmite $\delta^{18}\text{O}$ values. This paper takes a fundamentally different approach from the many excellent but sometimes contradictory modern studies by interrogating past responses. In fact, understanding how the peatland carbon cycle responds to drought on a long term, i.e, centennial or millennial timescale as discussed here, is even poorer. Our results provide not only a unique historical record but shed new light on a topic of intense contemporary debate.

The key point here is that evaporative effects likely differed between the sites – this is the key message for the main manuscript. However, we agree that our explanation is too perfunctory and have therefore moved it and expanded it (as above) in the Supplementary information.

Chen, J., Rao, Z., Liu, J., Huang, W., Feng, S., Dong, G., Hu, Y., Xu, Q., Chen F. On the timing of the East Asian summer monsoon maximum during the Holocene—Does the speleothem oxygen isotope record reflect monsoon rainfall variability? *Science China*

Earth Sciences, 59: 2328-2338 (2016).

Liu, Z., Wen, X., Brady, E.C., Otto-Bliesner, B., Yu, G., Lu, H., Cheng, H., Wang, Y., Zheng, W., Ding, Y., Edwards, R.L., Cheng, J., Liu, W., Yang, H. Chinese cave records and the East Asia summer monsoon. *Quat. Sci. Rev.* 83, 115-128.

Overall, we recognize that we have not been able to directly address several of Reviewer 2's requests for additional data. However, we have clarified several of the key issues. Moreover, by reframing our data as evidence for drier conditions in the peat – for which there is strong evidence – rather than drought, we hope we have mitigated some concerns. And we also think that by developing the interpretations of previous workers, we have provided a wider regional context for these new data without over-interpreting the current evidence. We agree that both higher-resolution $\delta^2\text{H}$ and radiocarbon data would be of great value, but argue that the fundamental relationships discussed here – between evidence for hydrological change and changes in carbon cycling – remain entirely valid.

Response to comments from Reviewer #3:

The authors have satisfactorily addressed my comments. I think the manuscript is much stronger with the wider implications written in very clearly.

Reply: We are grateful for the positive comments.

Reviewers' comments:

Reviewer #4 (Remarks to the Author):

The manuscript "Response of Carbon Cycle to Drier Conditions in the Mid-Holocene in Central China" provides a biomarker and compound-specific stable isotope record from a well-studied peatland in China. The data themselves do seem to support the conclusions drawn by the authors, but not as they are presented in the manuscript. The most convincing data to support the conclusions of the paper are hidden in the supplement.

The title of the manuscript mentions the impact of hydrology on the carbon cycle, but only TOC is presented. This is not a particularly useful measure of the carbon cycle as compared with the carbon accumulation rate (flux). The authors have the data to calculate this, and should show it.

The authors interpret δD of C29 as directly correlative with meteoric water. They contend that because the pollen-inferred vegetation is unchanging over the last 16ka, the apparent fractionation between water and lipids also does not change. However, pollen is not sufficient evidence to show that the vegetation producing the measured leaf waxes is not changing. Pollen provides a regional view of vegetation and is not necessarily indicative of the plants living in the peatland and especially not correlative with the plants contributing to the pool of alkanes in the peat. Even sphagnum spores are not an indicator of the presence or absence of sphagnum in the peat. Spores are only an indication of sexual reproduction, which is unrelated to its contribution of carbon or leaf waxes to the peat. Therefore, to say that the vegetation does not change based on pollen evidence is unconvincing. Macrofossil analysis, on the other hand, provides a much better view of the plants actually contributing to the peat and is the preferred paleovegetational proxy for interpreting leaf wax records. As the authors note, there are large differences in the apparent fractionation between water and waxes for different types of plants--greater than the 40 or so per mille variability in the C29 record reported here. Without better constraints on vegetation, δD of C29 cannot be directly interpreted as a meteoric water signal.

Why are the δD and $\delta^{13}C$ of sphagnum biomarkers (e.g., C23 n-alkane) not reported? Sphagnum water δD as represented by δD of C23 is much more sensitive to evaporative enrichment than that of vascular plants, like the C29 reported here. Also, $\delta^{13}C$ of sphagnum is sensitive to increased contribution of respired methane (as is mentioned here), yet $\delta^{13}C$ of C23 is also not reported. It seems that if the goal of the paper is to measure evaporative stress and increased methane oxidation, the authors have reported the isotope ratios of the wrong leaf wax compound. Perhaps low concentrations of C23 prevented the authors from reporting this measurement? The selected chromatogram in the supplement certainly suggests this, but the relative contribution of C23 and C29 is highly variable throughout the core, as a quick analysis of the alkane concentration data shows.

The fact that this C23/C29 ratio is variable also speaks to the previous point about changing contributions of different vegetation types throughout the core. While the authors conclude from pollen evidence (not shown) that vegetation is not changing, the alkane distributions suggest otherwise. In fact, the ACL record displayed in the supplement provides much more convincing evidence of dry conditions in the middle Holocene than the isotope data. ACL is much higher in the middle Holocene, indicating the reduced contribution of sphagnum to the peat as compared with the early and late Holocene. This reduced input by sphagnum is even clearer when the Paq ratio (Ficken et al., 2000) is applied to the data. Further, the age model (also only shown in the supplement) shows a clear decrease in accumulation rate during the time when contribution of sphagnum to the peat is low. Had this paper been written based on the alkane distribution data and the age model data from the supplement, rather than the tenuously interpreted δD data, it would have been much more convincing.

Overall, the data provided in this manuscript does strongly support the authors' broad conclusion

that hydrology impacts the carbon cycle in the peatland. Bafflingly, however, the most convincing data to support this conclusion is hidden in the supplement. This work can be an important contribution if the text is rewritten to emphasize the more conclusive aspects of the data. I look forward to reading a revised version.

Jonathan Nichols

Response to comments from Reviewer #4:

The manuscript “Response of Carbon Cycle to Drier Conditions in the Mid-Holocene in Central China” provides a biomarker and compound-specific stable isotope record from a well-studied peatland in China. The data themselves do seem to support the conclusions drawn by the authors, but not as they are presented in the manuscript. The most convincing data to support the conclusions of the paper are hidden in the supplement.

The title of the manuscript mentions the impact of hydrology on the carbon cycle, but only TOC is presented. This is not a particularly useful measure of the carbon cycle as compared with the carbon accumulation rate (flux). The authors have the data to calculate this, and should show it.

Reply: As suggested, the carbon accumulation rate was added to the new Fig. 5 in the revised text, with the calculation method being added to the experimental method section. During the drier interval in the mid-Holocene, the carbon accumulation rate is quite low (<10 g C/m²/yr), supporting our hypothesis that the drier conditions favor the peat decomposition. The two intervals with quite high carbon accumulation rate occurred at 10.1-9.5 ka and 3.4-3.0 ka, consistent with the relatively wetter conditions by our previous paleohydrological reconstructions (Huang et al., 2013; Xie et al., 2013). However, it should be noted that the carbon accumulation rate could be impacted by some other biotic/environmental factors in addition to the hydrological conditions. That is why the integration with other data, including our compound-specific carbon isotope data, is so powerful. In any case, we thank the reviewer for suggesting that we foreground these data.

Huang X., Xue J., Wang X., Meyers P.A., Huang J., Xie S. Paleoclimate influence on early diagenesis of plant triterpenes in the Dajiuhu Peatland, central China. *Geochimica et Cosmochimica Acta* 123, 106-119 (2013).

Xie, S., Evershed, R.P., Huang, X., Zhu, Z., Pancost, R., Meyers, P.A., Gong, L., Hu, C., Huang, J., Zhang, S., Gu, Y., Zhu, J. Concordant monsoon-driven postglacial hydrological changes in peat and stalagmite records and their impacts on prehistoric cultures in central China. *Geology* 41, 827-830 (2013).

The authors interpret δD of C29 as directly correlative with meteoric water. They contend that because the pollen-inferred vegetation is unchanging over the last 16ka, the apparent fractionation between water and lipids also does not change. However, pollen is not sufficient evidence to show that the vegetation producing the measured leaf waxes is not changing. Pollen provides a regional view of vegetation and is not necessarily indicative of the plants living in the peatland and especially not correlative with the plants contributing to the pool of alkanes in the peat. Even sphagnum spores are not an indicator of the presence or absence of sphagnum in the peat. Spores are only an indication of sexual reproduction, which is unrelated to its contribution of carbon or leaf waxes to the peat. Therefore, to say that the vegetation does not change based on pollen evidence is unconvincing. Macrofossil analysis, on the other hand, provides a much better view of the plants actually contributing to the peat and is the preferred paleovegetational proxy for interpreting leaf wax records. As the authors note, there are large differences in the apparent

fractionation between water and waxes for different types of plants--greater than the 40 or so per mille variability in the C₂₉ record reported here. Without better constraints on vegetation, δD of C₂₉ cannot be directly interpreted as a meteoric water signal.

Reply: Thank you for the good suggestion but the macrophyte data are not available in this study. In this subtropical peatland, the carbon accumulation rate, especially during the mid-Holocene, is quite low, and the humification is rather high (Please see the lowest panel in the Fig. 6 of Huang et al., 2015). Such a condition makes the poor preservation of the macrophytes. The reviewer is correct that there is uncertainty to evaluate the impact of the vegetation change on the variations of δD of C₂₉ *n*-alkane due to the limitations of pollen and the lack of macrophyte evidence; however, a line of evidence shown below does not support the vegetation is an important factor to cause the large fluctuations of the δD values of C₂₉ *n*-alkane in the drier mid-Holocene intervals.

(1) As suggested by the reviewer below, we can infer some vegetation changes using biomarker distributions. To the first order, they exhibit reinforcing patterns – dry conditions indicated by δD values are also indicated by *n*-alkane distributions. However, there no correlation between individual δD and vegetation biomarker data. For example, during the dry interval, biomarkers indicate a prolonged vegetation change, but δD values indicate episodic events; i.e. being both high and low despite no vegetation change.

(2) Investigations on the modern vegetation (Luo et al., 2015) and the *n*-alkane distribution in surficial peat samples (Huang et al., 2011, 2014) as well as the wetland pollen data (Zhu et al., 2010) support the dominant contribution of C₃ herb plants after 9 ka BP, and for the C₃ herb plants, there is a narrow range of $\epsilon_{\text{lipid/water}}$ values (Sachse et al., 2012). Changes of C₃ herb plants could not explain the observed δD of C₂₉ *n*-alkane, especially after 9 ka.

In contrast, the interpretation of drought-induced evaporation on the δD of C₂₉ *n*-alkane is supported by the vertical profiles of pore water δD . More importantly, the drier condition inferred by the relatively enriched δD_{29} values was consistent with other independent proxies (hopanoid flux, Xie et al., 2013; the average aromatic ratio, Huang et al., 2013) in the same peatland, as well as with other proxies (magnetic ratio; Zhu et al., 2017) at the same region. We have added some further caveats to this section to convey why we are confident that meteoric water is the primary control on δD shifts, but also to acknowledge the concerns of the reviewer.

Huang, X., Wang, C., Zhang, J., Wiesenberg, G.L.B., Zhang, Z., Xie, S. Comparison of free lipid compositions between roots and leaves of plants in the Dajiuhu Peatland, central China. *Geochem. J.* 45, 365-373 (2011).

Huang X., Meyers P.A., Xue J., Gong L., Wang X., Xie S. Environmental factors affecting the low temperature isomerization of homohopanes in acidic peat deposits, central China. *Geochimica et Cosmochimica Acta* 154, 212-228 (2015).

Huang, X., Xue, J., Wang, X., Meyers, P.A., Gong, L., Liu, Q., Qin, Y., Wang, H. Hydrologic influence on $\delta^{13}C$ variations in long-chain *n*-alkanes in the Dajiuhu peatland, central China. *Org. Geochem.* 69, 114-119 (2014).

Luo, T., Lun, Z., Gu, Y., et al., 2015. Plant community survey and ecological protection of Dajiuhu Wetlands in Shennongjia area. *Wetland Science*, 13(2): 153-160 (in Chinese)

with English abstract).

- Sachse, D., Billault, I., Bowen, G. J., Chikaraishi, Y., Dawson, T. E., Feakins, S. J., Freeman, K. H., Magill, C. R., McInerney, F. A., van Der Meer, M. T. J., Polissar, P., Robins, R. J., Sachs, J. P., Schmidt, H.-L., Sessions, A. L., White, J. W. C., West, J. B., Kahmen A. Molecular paleohydrology: interpreting the hydrogen-isotopic composition of lipid biomarkers from photosynthesizing organisms. *Annu. Rev. Earth Planet. Sci.* 40, 221–249 (2012).
- Xie, S., Evershed, R.P., Huang, X., Zhu, Z., Pancost, R., Meyers, P.A., Gong, L., Hu, C., Huang, J., Zhang, S., Gu, Y., Zhu, J. Concordant monsoon-driven postglacial hydrological changes in peat and stalagmite records and their impacts on prehistoric cultures in central China. *Geology* 41, 827-830 (2013).
- Zhu, C., Ma, C., Yu, S.-Y., Tang, L., Zhang, W., Lu, X. A detailed pollen record of vegetation and climate changes in Central China during the past 16000 years. *Boreas* 39, 69-76 (2010).
- Zhu, Z., Feinberg, J.M., Xie, S., Bourne, M.D., Huang, C., Hu, C., Cheng, H. Holocene ENSO-related cyclic storms recorded by magnetic minerals in speleothem of central China. *P. Natl. Acad. Sci. USA* 114, 852-857 (2017).

Why are the δD and $\delta^{13}C$ of sphagnum biomarkers (e.g., C_{23} n-alkane) not reported? Sphagnum water δD as represented by δD of C_{23} is much more sensitive to evaporative enrichment than that of vascular plants, like the C_{29} reported here. Also, $\delta^{13}C$ of sphagnum is sensitive to increased contribution of respired methane (as is mentioned here), yet $\delta^{13}C$ of C_{23} is also not reported. It seems that if the goal of the paper is to measure evaporative stress and increased methane oxidation, the authors have reported the isotope ratios of the wrong leaf wax compound. Perhaps low concentrations of C_{23} prevented the authors from reporting this measurement? The selected chromatogram in the supplement certainly suggests this, but the relative contribution of C_{23} and C_{29} is highly variable throughout the core, as a quick analysis of the alkane concentration data shows.

The fact that this C_{23}/C_{29} ratio is variable also speaks to the previous point about changing contributions of different vegetation types throughout the core. While the authors conclude from pollen evidence (not shown) that vegetation is not changing, the alkane distributions suggest otherwise. In fact, the ACL record displayed in the supplement provides much more convincing evidence of dry conditions in the middle Holocene than the isotope data. ACL is much higher in the middle Holocene, indicating the reduced contribution of sphagnum to the peat as compared with the early and late Holocene. This reduced input by sphagnum is even clearer when the Paq ratio (Ficken et al., 2000) is applied to the data. Further, the age model (also only shown in the supplement) shows a clear decrease in accumulation rate during the time when contribution of sphagnum to the peat is low. Had this paper been written based on the alkane distribution data and the age model data from the supplement, rather than the tenuously interpreted δD data, it would have been much more convincing.

Reply: Thanks for the good suggestion to the addition of the *sphagnum*-associated proxies in the discussion. As mentioned by the reviewer, the low concentration of C_{23} n-alkane relative to longer homologues (e.g. C_{29} and C_{31}) makes it difficult to precisely measure the δD and

$\delta^{13}\text{C}$ values of C_{23} . However, we did add some data points of the $\delta^{13}\text{C}$ of C_{23} *n*-alkane in the new Fig. 5 when the isotope data can be available (But the incompleteness of this record means we cannot use it to infer temporal trends as suggested). The $\text{C}_{23}/\text{C}_{29}$ ratios, together with ACL and Paq records, are now presented as a new Fig. 3 in the revised text; as suggested by the reviewer they add additional evidence for dry conditions during the mid-Holocene. We disagree that this is stronger evidence; but certainly, the confluence of δD values, *n*-alkane ratios and hopanoid abundances is strong evidence for drier conditions.

Following the comments from the reviewer, we have added the evidence of alkane ratios (ACL, Paq, and $\text{C}_{23}/\text{C}_{29}$) to the revised text to support the conclusion of a drier mid-Holocene, though lines of evidence published previously do support this conclusion.

Overall, the data provided in this manuscript does strongly support the authors' broad conclusion that hydrology impacts the carbon cycle in the peatland. Bafflingly, however, the most convincing data to support this conclusion is hidden in the supplement. This work can be an important contribution if the text is rewritten to emphasize the more conclusive aspects of the data. I look forward to reading a revised version.

Reply: Thanks for the comments. As suggested, we have moved some important information, such as the alkane ratios and the $\delta^{13}\text{C}$ values, from the supplementary materials to the main text. In addition, the peat carbon accumulation rate was added in the text and in the figure. These new data can further consolidate our discussion on the biogeochemical responses of peatland ecosystem to the drier conditions during the mid-Holocene in central China. We do disagree on what is the most compelling line of evidence, but that is a rather moot point since all of the evidence yields a compelling and coherent interpretation. We thank the reviewer for urging us to include this wider variety of evidence in the main manuscript.

REVIEWERS' COMMENTS:

Reviewer #4 (Remarks to the Author):

The authors have sufficiently addressed the points raised in my previous review. The manuscript is now ready for publication. However, I would also note that while the authors do provide the data from this investigation in the supplement, it would be best if the data were submitted to a database so that it can be more easily accessed. The Neotoma Paleocology Database, for example, has just begun accepting biomarker concentration and compound-specific stable isotope data in addition to pollen, macrofossil, and carbon content data. The Pangaea database would also be an appropriate place to deposit data.

REVIEWERS' COMMENTS:

Reviewer #4 (Remarks to the Author):

The authors have sufficiently addressed the points raised in my previous review. The manuscript is now ready for publication. However, I would also note that while the authors do provide the data from this investigation in the supplement, it would be best if the data were submitted to a database so that it can be more easily accessed. The Neotoma Paleoecology Database, for example, has just begun accepting biomarker concentration and compound-specific stable isotope data in addition to pollen, macrofossil, and carbon content data. The Pangaea database would also be an appropriate place to deposit data.

Reply: Thanks for the positive comment. We have sent the original data to the Reviewer #4 (Dr. Jon Nichols), who is the lead data steward for organic geochemistry of the Neotoma Paleocology Database. Please see the attached email screenshot:

Because the uploaded data need to be processed for some time by the agent of the database, we could not provide the repository information recently. Anyway, all these data are included in the supplementary material which we submit to the journal, and the data availability statements are added in the main text.